# De novo design of luciferases using deep learning

Andy Hsien-Wei Yeh[1,2,3,7 ✉], Christoffer Norn[1,2,7], Yakov Kipnis[1,2,4], Doug Tischer[1,2], Samuel J. Pellock[1,2], Declan Evans[5], Pengchen Ma[5,6], Gyu Rie Lee[1,2], Jason Z. Zhang[1,2], Ivan Anishchenko[1,2], Brian Coventry[1,2,4], Longxing Cao[1,2], Justas Dauparas[1,2], Samer Halabiya[2], Michelle DeWitt[2], Lauren Carter[2], K. N. Houk[5] & David Baker[1,2,4 ✉]

De novo enzyme design has sought to introduce active sites and substrate-binding pockets that are predicted to catalyse a reaction of interest into geometrically compatible native scaffolds[1,2], but has been limited by a lack of suitable protein structures and the complexity of native protein sequence–structure relationships. Here we describe a deep-learning-based 'family-wide hallucination' approach that generates large numbers of idealized protein structures containing diverse pocket shapes and designed sequences that encode them. We use these scaffolds to design artificial luciferases that selectively catalyse the oxidative chemiluminescence of the synthetic luciferin substrates diphenylterazine[3] and 2-deoxycoelenterazine. The designed active sites position an arginine guanidinium group adjacent to an anion that develops during the reaction in a binding pocket with high shape complementarity. For both luciferin substrates, we obtain designed luciferases with high selectivity; the most active of these is a small (13.9 kDa) and thermostable (with a melting temperature higher than 95 °C) enzyme that has a catalytic efficiency on diphenylterazine ($k_{cat}/K_m = 10^6\,M^{-1}\,s^{-1}$) comparable to that of native luciferases, but a much higher substrate specificity. The creation of highly active and specific biocatalysts from scratch with broad applications in biomedicine is a key milestone for computational enzyme design, and our approach should enable generation of a wide range of luciferases and other enzymes.

Bioluminescent light produced by the enzymatic oxidation of a luciferin substrate by luciferases is widely used for bioassays and imaging in biomedical research. Because no excitation light source is needed, luminescent photons are produced in the dark; this results in higher sensitivity than fluorescence imaging in live animal models and in biological samples in which autofluorescence or phototoxicity is a concern[4,5]. However, the development of luciferases as molecular probes has lagged behind that of well-developed fluorescent protein toolkits for a number of reasons: (i) very few native luciferases have been identified[6,7]; (ii) many of those that have been identified require multiple disulfide bonds to stabilize the structure and are therefore prone to misfolding in mammalian cells[8]; (iii) most native luciferases do not recognize synthetic luciferins with more desirable photophysical properties[9]; and (iv) multiplexed imaging to follow multiple processes in parallel using mutually orthogonal luciferase–luciferin pairs has been limited by the low substrate specificity of native luciferases[10,11].

We sought to use de novo protein design to create luciferases that are small, highly stable, well-expressed in cells, specific for one substrate and need no cofactors to function. We chose a synthetic luciferin, diphenylterazine (DTZ), as the target substrate because of its high quantum yield, red-shifted emission[3], favourable in vivo pharmacokinetics[12,13] and lack of required cofactors for light emission. Previous computational enzyme design efforts have primarily repurposed native protein scaffolds in the Protein Data Bank (PDB)[1,2], but there are few native structures with binding pockets appropriate for DTZ, and the effects of sequence changes on native proteins can be unpredictable (designed helical bundles have also been used as enzyme scaffolds[14–16], but these are limited in number and most do not have pockets that are suitable for DTZ binding). To circumvent these limitations, we set out to generate large numbers of small and stable protein scaffolds with pockets of the appropriate size and shape for DTZ, and with clear sequence–structure relationships to facilitate subsequent active-site incorporation. To identify protein folds that are capable of hosting such pockets, we first docked DTZ into 4,000 native small-molecule-binding proteins. We found that many nuclear transport factor 2 (NTF2)-like folds have binding pockets with appropriate shape complementarity and size for DTZ placement (pink dashes in Fig. 1e), and hence selected the NTF2-like superfamily as the target topology.

[1]Department of Biochemistry, University of Washington, Seattle, WA, USA. [2]Institute for Protein Design, University of Washington, Seattle, WA, USA. [3]Department of Biomolecular Engineering, University of California, Santa Cruz, CA, USA. [4]Howard Hughes Medical Institute, University of Washington, Seattle, WA, USA. [5]Department of Chemistry and Biochemistry, University of California, Los Angeles, CA, USA. [6]School of Chemistry, Xi'an Key Laboratory of Sustainable Energy Materials Chemistry, MOE Key Laboratory for Nonequilibrium Synthesis and Modulation of Condensed Matter, Xi'an Jiaotong University, Xi'an, China. [7]These authors contributed equally: Andy Hsien-Wei Yeh, Christoffer Norn. ✉e-mail: hsyeh@ucsc.edu; dabaker@uw.edu

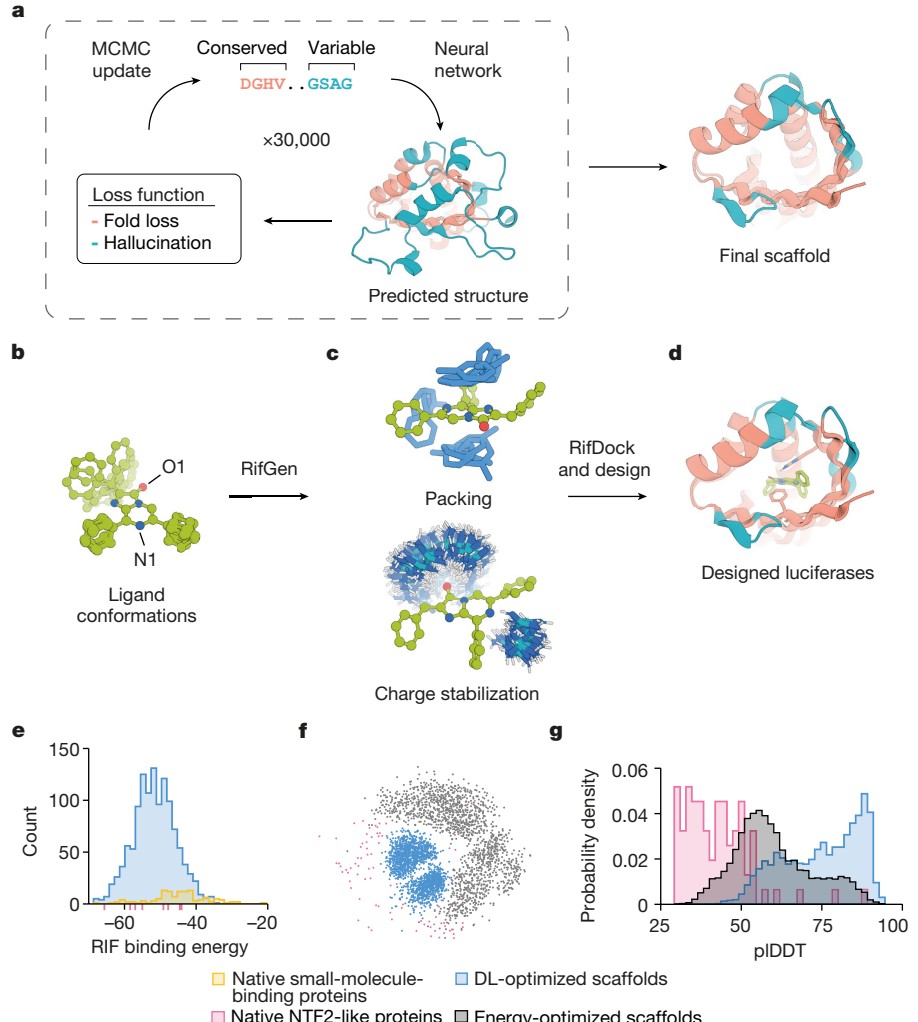

**Fig. 1 | Generation of idealized scaffolds and computational design of de novo luciferases. a**, Family-wide hallucination. Sequences encoding proteins with the desired topology are optimized by Markov chain Monte Carlo (MCMC) sampling with a multicomponent loss function. Structurally conserved regions (peach) are evaluated on the basis of consistency with input residue–residue distance and orientation distributions obtained from 85 experimental structures of NTF2-like proteins, whereas variable non-ideal regions (teal) are evaluated on the basis of the confidence of predicted inter-residue geometries calculated as the KL divergence between network predictions and the background distribution. The sequence-space MCMC sampling incorporates both sequence changes and insertions and deletions (see Supplementary Methods) to guide the hallucinated sequence towards encoding structures with the desired folds. Hydrogen-bonding networks are incorporated into the designed structures to increase structural specificity. **b–d**, The design of luciferase active sites. **b**, Generation of luciferase substrate (DTZ) conformers. **c**, Generation of a Rotamer Interaction Field (RIF) to stabilize anionic DTZ and form hydrophobic packing interactions. **d**, Docking of the RIF into the hallucinated scaffolds, and optimization of substrate–scaffold interactions using position-specific score matrices (PSSM)-biased sequence design. **e**, Selection of the NTF2 topology. The RIF was docked into 4,000 native small-molecule-binding proteins, excluding proteins that bind the luciferin substrate using more than five loop residues. Most of the top hits were from the NTF2-like protein superfamily (pink dashes). Using the family-wide hallucination scaffold generation protocol, we generated 1,615 scaffolds and found that these yielded better predicted RIF binding energies than the native proteins. **f,g**, Our DL-optimized scaffolds sample more within the space of the native structures (**f**) and have stronger sequence-to-structure relationships (more confident Alphafold2 structure predictions) (**g**) than native or previous non-deep-learning energy-optimized scaffolds.

## Family-wide hallucination

Native NTF2 structures have a range of pocket sizes and shapes but also contain features that are not ideal, such as long loops that compromise stability. To create large numbers of ideal NTF2-like structures, we developed a deep-learning-based 'family-wide hallucination' approach that integrates unconstrained de novo design[17,18] and Rosetta sequence-design approaches[19] to enable the generation of an essentially unlimited number of proteins that have a desired fold (Fig. 1a). The family-wide hallucination approach used the de novo sequence and structure discovery capability of unconstrained protein hallucination[17,18] for loop and variable regions, and structure-guided sequence optimization for core regions. We used the trRosetta structure prediction neural network[20], which is effective in identifying experimentally successful de-novo-designed proteins and hallucinating new globular proteins of diverse topologies. Starting from the sequences of 2,000 naturally occurring NTF2s, we carried out Monte Carlo searches in sequence space, at each step making a sequence change and predicting the structure using trRosetta. As the loss function guiding search, we used the confidence of the neural network in the predicted structure (as in our previous free hallucination study) supplemented with a topology-specific loss function over core residue pair geometries (see Supplementary Methods); in the loop regions, we also allowed the number of residues to vary,

which resulted in short near ideal loops. To further encode structural specificity, we incorporated buried, long-range hydrogen-bonding networks. The resulting 1,615 family-wide hallucinated NTF2 scaffolds provided more shape-complementary binding pockets for DTZ than did native small-molecule-binding proteins (Fig. 1e). This method samples protein backbones that are closer to native NTF2-like proteins (Fig. 1f) and that have better scaffold quality metrics than those produced in a previous non-deep-learning energy-based approach[21] (Fig. 1g).

## De novo design of luciferases for DTZ

Computational enzyme design generally starts from an ideal active site or theozyme consisting of protein functional groups surrounding the reaction transition state that is then matched into a set of existing scaffolds[1,2]. However, the detailed catalytic geometry of native marine luciferases is not well understood because only a handful of apo structures and no holo structures with luciferin substrates have been solved (at the time of this study)[22–24]. Both quantum chemistry calculations[25,26] and experimental data[27,28] suggest that the chemiluminescent reaction proceeds through an anionic species and that the polarity of the surroundings can substantially alter the free energy of the subsequent single-electron transfer (SET) process with triplet molecular oxygen ($^3O_2$). Guided by these data (Extended Data Fig. 1), we sought to design a shape-complementary catalytic site that stabilizes the anionic state of DTZ and lowers the SET energy barrier, assuming that the downstream dioxetane light emitter thermolysis steps are spontaneous. To stabilize the anionic state, we focused on the placement of the positively charged guanidinium group of an arginine residue to stabilize the developing negative charge on the imidazopyrazinone group.

To computationally design such active sites into large numbers of hallucinated NTF2 scaffolds, we first generated an ensemble of anionic DTZ conformers (Fig. 1b). Next, around each conformer, we used the RifGen method[29,30] to enumerate rotamer interaction fields (RIFs) on three-dimensional grids consisting of millions of placements of amino acid side chains making hydrogen-bonding and nonpolar interactions with DTZ (Fig. 1c). An arginine guanidinium group was placed adjacent to the N1 atom of the imidazopyrazinone group to stabilize the negative charge. RifDock was then used to dock each DTZ conformer and associated RIF in the central cavity of each scaffold to maximize protein–DTZ interactions. An average of eight side-chain rotamers, including an arginine residue to stabilize the anionic imidazopyrazinone core, were positioned in each pocket (Supplementary Fig. 2a). For the top 50,000 docks with the most favourable side chain–DTZ interactions, we optimized the remainder of the sequence using RosettaDesign (Fig. 1d) for high-affinity binding to DTZ with a bias towards the naturally observed sequence variation to ensure foldability. During the design process, pre-defined hydrogen-bond networks (HBNets) in the scaffolds were kept intact for structural specificity and stability, and interactions of these HBNet side chains with DTZ were explicitly required in the Rif-Dock step to ensure the preorganization of residues that are essential for catalysis. In the first sequence-design step, the identities of all RIF and HBNet residues were kept fixed, and the surrounding residues were optimized to hold the side chain–DTZ interactions in place and maintain structural specificity. In the second sequence-design step, the RIF residue identities (except the arginine) were also allowed to vary, as Rosetta can identify apolar and aromatic packing interactions that were missed in the RIF owing to binning effects. During sequence design, the scaffold backbone, side chains and DTZ substrate were allowed to relax in Cartesian space. After sequence optimization, the designs were filtered on the basis of ligand-binding energy, protein–ligand hydrogen bonds, shape complementarity and contact molecular surface, and 7,648 designs were selected and ordered as pooled oligos for experimental screening.

## Identification of active luciferases

Oligonucleotides encoding the two halves of each design were assembled into full-length genes and cloned into an *Escherichia coli* expression vector (see Supplementary Methods). A colony-based screening method was used to directly image active luciferase colonies from the library and the activities of selected clones were confirmed using a 96-well plate expression (Extended Data Fig. 2). Three active designs were identified; we refer to the most active of these as LuxSit (from the Latin lux sit, 'let light exist'), which at 117 residues (13.9 kDa) is, to our knowledge, smaller than any previously described luciferase. Biochemical analysis, including SDS–PAGE and size-exclusion chromatography (Fig. 2a,b and Extended Data Fig. 3), indicated that LuxSit is highly expressed in *E. coli*, soluble and monomeric. Circular dichroism (CD) spectroscopy showed a strong far-ultraviolet CD signature, suggesting an organized α-β structure. CD melting experiments showed that the protein is not fully unfolded at 95 °C, and that the full structure is regained when the temperature is dropped (Fig. 2c). Incubation of LuxSit with DTZ resulted in luminescence with an emission peak at around 480 nm (Fig. 2d), consistent with the DTZ chemiluminescence spectrum. Although we were not able to determine the crystal structure of LuxSit, the structure predicted by AlphaFold2 (ref. [31]) is very close to the design model at the backbone level (root-mean-square deviation (RMSD) = 1.35 Å) and over the side chains interacting with the substrate (Fig. 2e). The designed LuxSit active site contains Tyr14–His98 and Asp18–Arg65 dyads, with the imidazole nitrogen atoms of His98 making hydrogen-bond interactions with Tyr14 and the O1 atom of DTZ (Fig. 2f). The centre of the Arg65 guanidinium cation is 4.2 Å from the N1 atom of DTZ and Asp18 forms a bidentate hydrogen bond to the guanidinium group and backbone N–H of Arg65 (Fig. 2g).

## De novo design of luciferases for h-CTZ

We next sought to apply the knowledge gained from designing LuxSit to create 2-deoxycoelenterazine (h-CTZ)-specific luciferases. Because the molecular shape of h-CTZ is different from that of DTZ, we created an additional set of NTF2 superfamily scaffolds (see Supplementary Methods) with matching pocket shapes and high model confidence (AlphaFold2-predicted local-distance difference test (pLDDT) > 92). We then installed catalytic sites in these scaffolds and designed the first shell-protein side chain–h-CTZ interactions using the histidine and arginine substrate interaction geometries that were most successful in the first round for DTZ. To design the remainder of the sequence, we used ProteinMPNN[32], which can result in better stability, solubility and accuracy than RosettaDesign. After filtering on the basis of the AlphaFold2-predicted pLDDT, Cα RMSD, contact molecular surface and Rosetta-computed binding energies (see Supplementary Methods), we selected and experimentally expressed 46 designs in *E. coli* and identified 2 (HTZ3-D2 and HTZ3-G4) that had luciferase activity with the h-CTZ luciferin substrate. Both designs were highly soluble, monodisperse and monomeric, and the luciferase activities were of the same order of magnitude as LuxSit (Extended Data Fig. 4). The success rate increased from 3/7,648 to 2/46 sequences in the second round, probably owing to the knowledge of active-site geometry from the first round and the increased robustness of the ProteinMPNN method of sequence design.

## Optimization of luciferase activity

To better understand the contributions to the catalysis of LuxSit, the most active of our designs, we constructed a site-saturation mutagenesis (SSM) library in which each residue in the substrate-binding pocket was mutated to every other amino acid one at a time (see Supplementary Methods), and determined the effect of each mutation on luciferase activity. Figure 2f–i shows the amino acid preferences at key positions.

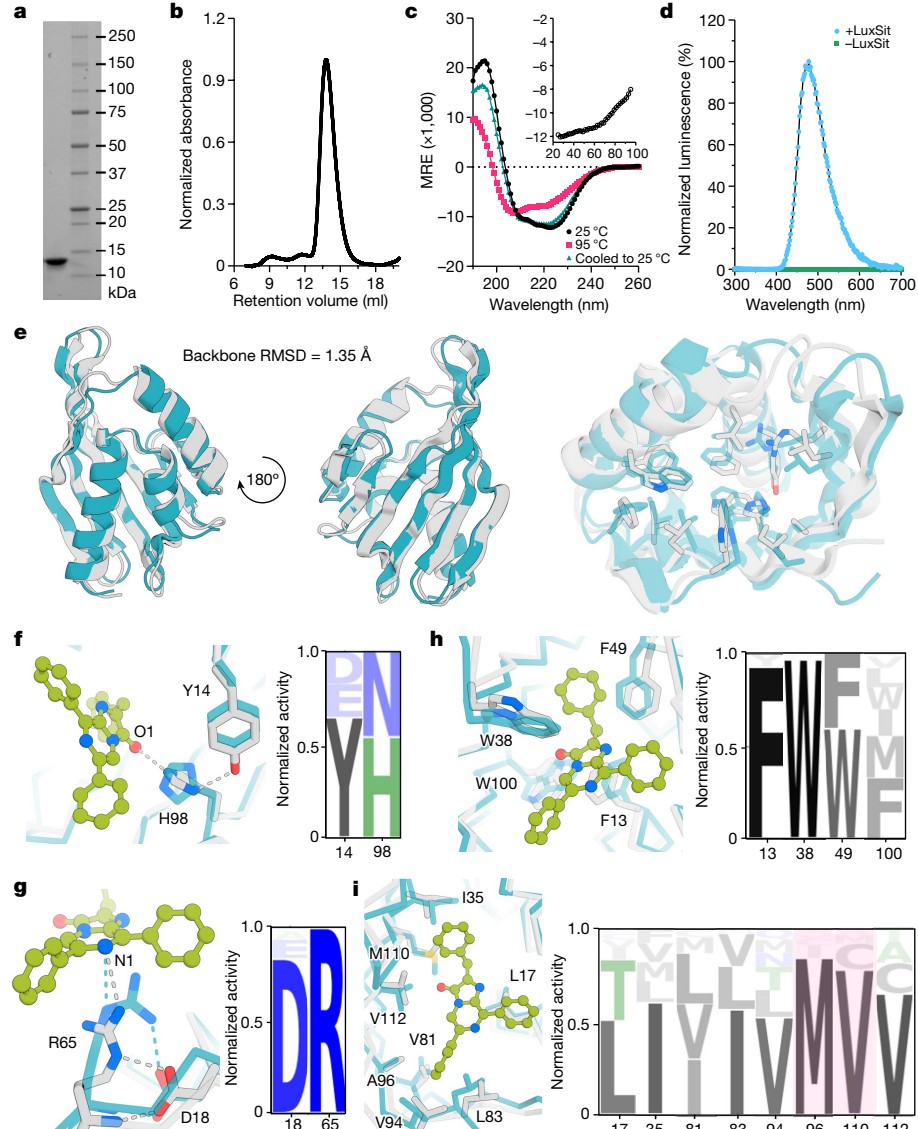

**Fig. 2 | Biophysical characterization of LuxSit. a,** Coomassie-stained SDS–PAGE of purified recombinant LuxSit from *E. coli* (for gel source data, see Supplementary Fig. 1). **b,** Size-exclusion chromatography of purified LuxSit suggests monodispersed and monomeric properties. **c,** Far-ultraviolet CD spectra at 25 °C (black), 95 °C (red) and cooled back to 25 °C (green). Insert, CD melting curve of LuxSit at 220 nm. MRE, molar residue ellipticity. **d,** Luminescence emission spectra of DTZ in the presence (blue) and absence (green) of LuxSit. **e,** Structural alignment of the design model (blue) and AlphaFold2-predicted model (grey), which are in close agreement at both the backbone (left) and the side-chain (right) level. **f–i,** Site-saturation mutagenesis of substrate-interacting residues. Magnified views (left) of designed (blue) and AlphaFold2 (grey) models at the side-chain level, illustrating the designed enzyme–substrate interactions of Tyr14–His98 core HBNet (**f**), Asp18–Arg65 dyad (**g**), π-stacking (**h**) and hydrophobic packing (**i**) residues. Sequence profiles (right) are scaled by the activities of different sequence variants: (activity for the indicated amino acid)/(sum of activities over all tested amino acids at the indicated position). A96M and M110V substitutions with increased activity are highlighted in pink.

Arg65 is highly conserved (Fig. 2g), and its dyad partner Asp18 can only be mutated to Glu (which reduces activity), suggesting that the carboxylate–Arg65 hydrogen bond is important for luciferase activity. In the Tyr14–His98 dyad (Fig. 2f), Tyr14 can be substituted with Asp and Glu, and His98 can be replaced with Asn. As all active variants had hydrogen-bond donors and acceptors at these positions, the dyads might help to mediate the electron and proton transfer required for luminescence. Hydrophobic (Fig. 2i) and π-stacking (Fig. 2h) residues at the binding interface tolerate other aromatic or aliphatic substitutions and generally prefer the amino acid in the original design, consistent with model-based affinity predictions of mutational effects (Extended Data Fig. 5). The A96M and M110V mutants (highlighted in pink) increase activity by 16-fold and 19-fold, respectively, over LuxSit (Supplementary

Table 1). Optimization guided by these results yielded LuxSit-f (A96M/M110V), with a flash-type emission kinetic, and LuxSit-i (R60S/A96L/M110V), with a photon flux more than 100-fold higher than that of LuxSit (Extended Data Fig. 6). Overall, the active-site-saturation mutagenesis results support the design model, with the Tyr14–His98 and Asp18–Arg65 dyads having key roles in catalysis and the substrate-binding pocket largely conserved.

The most active catalysts, LuxSit-i (Extended Data Fig. 3b,e,h) and LuxSit-f (Extended Data Fig. 3c,f,i), were both expressed solubly in *E. coli* at high levels and are monomeric (some dimerization was observed at the high protein concentration; Extended Data Fig. 3l) and thermostable (Extended Data Fig. 3j,k). Similar to native luciferases that use CTZ, the apparent Michaelis constants ($K_m$) of both LuxSit-i and

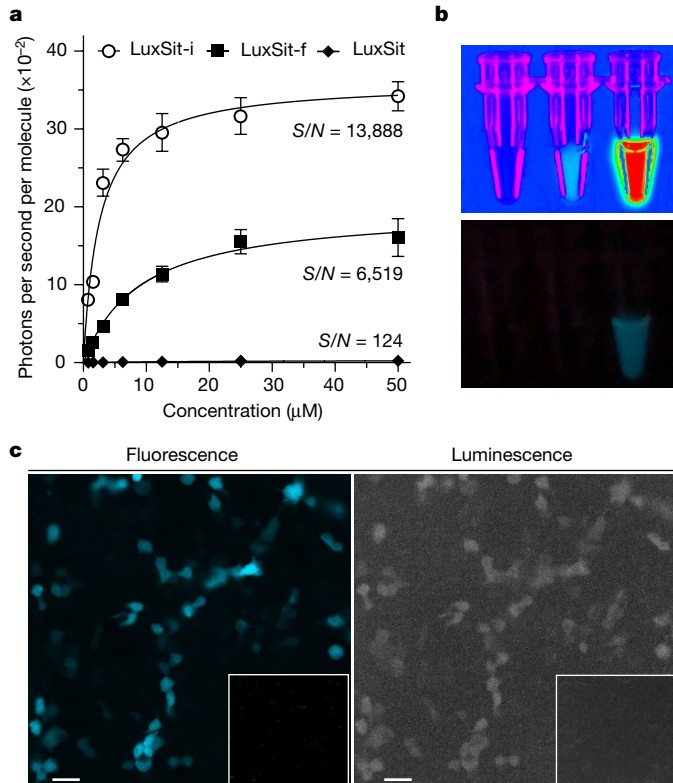

**Fig. 3 | Characterization of de novo luciferase activity in vitro and in human cells. a**, Substrate concentration dependence of LuxSit, LuxSit-f and LuxSit-i activity. Numbers indicate the signal-to-background ($S$/$N$) ratio at $V_{max}$ (photon s$^{-1}$ molecule$^{-1}$). Data are mean ± s.d. ($n$ = 3). **b**, Luminescence images acquired by a BioRad Imager (top) or an Apple iPhone 8 camera (bottom). Tubes from left to right: DTZ only; DTZ plus 100 nM purified LuxSit; and DTZ plus 100 nM purified LuxSit-i, showing the high efficiency of photon production. **c**, Fluorescence and luminescence microscopic images of live HEK293T cells transiently expressing LuxSit-i-mTagBFP2; LuxSit-i activity can be detected at single-cell resolution. Left, fluorescence channel representing the mTagBFP2 signal. Right, total luminescence photons were collected during a course of a 10-s exposure without excitation light, immediately after adding 25 μM DTZ. Insets, negative control, untransfected cells with DTZ. Scale bars, 20 μm; 40× magnification.

LuxSit-f are in the low-micromolar range (Fig. 3a) and the luminescent signal decays over time owing to fast catalytic turnover (Extended Data Fig. 7a). LuxSit-i is a very efficient enzyme, with a catalytic efficiency ($k_{cat}$/$K_m$) of $10^6$ M$^{-1}$ s$^{-1}$. The luminescence signal is readily visible to the naked eye (Fig. 3b), and the photon flux (photons per second) is 38% greater than that of the native *Renilla reniformis* luciferase (RLuc) (Supplementary Table 2). The DTZ luminescent reaction catalysed by LuxSit-i is pH-dependent (Extended Data Fig. 7b), consistent with the proposed mechanism. We used a combination of density functional theory (DFT) calculations and molecular dynamics (MD) simulations to investigate the basis for LuxSit activity in more detail; the results support the anion-stabilization mechanism (Extended Data Fig. 8a and Supplementary Fig. 3a) and suggest that LuxSit-i provides better DTZ transition-state charge stabilization than LuxSit (Extended Data Fig. 8b).

## Cell imaging and multiplexed bioassay

As luciferases are commonly used genetic tags and reporters for cell biological studies, we evaluated the expression and function of LuxSit-i in live mammalian cells. HEK293T cells expressing LuxSit-i-mTagBFP2

showed DTZ-specific luminescence (Fig. 3c), which was maintained after targeting of LuxSit-i-mTagBFP2 to the nucleus, membrane and mitochondria (Extended Data Fig. 9). Native and previously engineered luciferases are quite promiscuous, with activity on many luciferin substrates (Fig. 4ac and Supplementary Fig. 4); this is possibly a result of their large and open pockets (a luciferase with high specificity to one luciferin substrate has been difficult to control even with extensive directed evolution[33,34]). By contrast, LuxSit-i exhibited exquisite specificity for its target luciferin, with 50-fold selectivity for DTZ over bis-CTZ (which differs only in one benzylic carbon; MD simulations suggest that this arises from greater transition-state shape complementarity (Extended Data Fig. 8b,c and Supplementary Fig. 3b,c)), 28-fold selectivity over 8pyDTZ (differing only in one nitrogen atom) and more than 100-fold selectivity over other luciferin substrates (Fig. 4b). One of our active design for h-CTZ (HTZ3-G4) was also highly specific for its target substrate (Fig. 4c and Extended Data Fig. 4d). Overall, the specificity of our designed luciferases is much greater than that of native luciferases[35,36] or previously engineered luciferases[37] (Supplementary Table 5).

We reasoned that the high substrate specificity of LuxSit-i could allow the multiplexing of luminescent reporters through substrate-specific or spectrally resolved luminescent signals (Fig. 4d and Extended Data Fig. 10a,b). To investigate this possibility, we placed LuxSit-i downstream of the NF-κB response element and RLuc downstream of the cAMP response element (Fig. 4e). The addition of activators (TNF) of the NF-κB signaling pathway resulted in luminescence when cells were incubated with DTZ, while the luminescence of PP-CTZ (the substrate of RLuc) was observed only when the cAMP–PKA pathway was activated (Fig. 4f). Because DTZ and PP-CTZ emit luminescence at different wavelengths, they can in principle be combined and the two signals can be deconvoluted through spectral analysis. Indeed, we observed that activating the NF-κB signaling resulted in luminescence at the DTZ wavelength, while the addition of cAMP–PKA pathway activators (FSK) generated luminescence at the PP-CTZ wavelength, allowing us to simultaneously assess the activation of the two signaling pathways in the same sample with either cell lysates (Fig. 4g) or intact HEK293T cells (Extended Data Fig. 10c–e) by providing both substrates together. Thus, the high substrate specificity of LuxSit-i enables multiplexed reporting of diverse cellular responses.

## Conclusion

Computational enzyme design has been constrained by the number of available scaffolds, which limits the extent to which catalytic configurations and enzyme–substrate shape complementarity can be achieved[14–16]. The use of deep learning to produce large numbers of de-novo-designed scaffolds here eliminates this restriction, and the more accurate RoseTTAfold (ref. [38]) and AlphaFold2 (ref. [31]) should enable protein scaffolds to be generated even more effectively through family-wide hallucination and other approaches[18,39]. The diversity of shapes and sizes of scaffold pockets enabled us to consider a range of catalytic geometries and to maximize reaction intermediate–enzyme shape complementarity; to our knowledge, no native luciferases have folds similar to LuxSit, and the enzyme has high specificity for a fully synthetic luciferin substrate that does not exist in nature. With the incorporation of three substitutions that provide a more complementary pocket to stabilize the transition state, LuxSit-i has higher activity than any previous de-novo-designed enzyme, with a $k_{cat}$/$K_m$ ($10^6$ M$^{-1}$ s$^{-1}$) in the range of native luciferases. This is a notable advance for computational enzyme design, as tens of rounds of directed evolution were required to obtain catalytic efficiencies in this range for a designed retroaldolase, and the structure was remodelled considerably[40]; by contrast, the predicted differences in ligand–side-chain interactions between LuxSit and LuxSit-i are very subtle (Supplementary Fig. 2b; achieving such high activities directly from the computer remains a

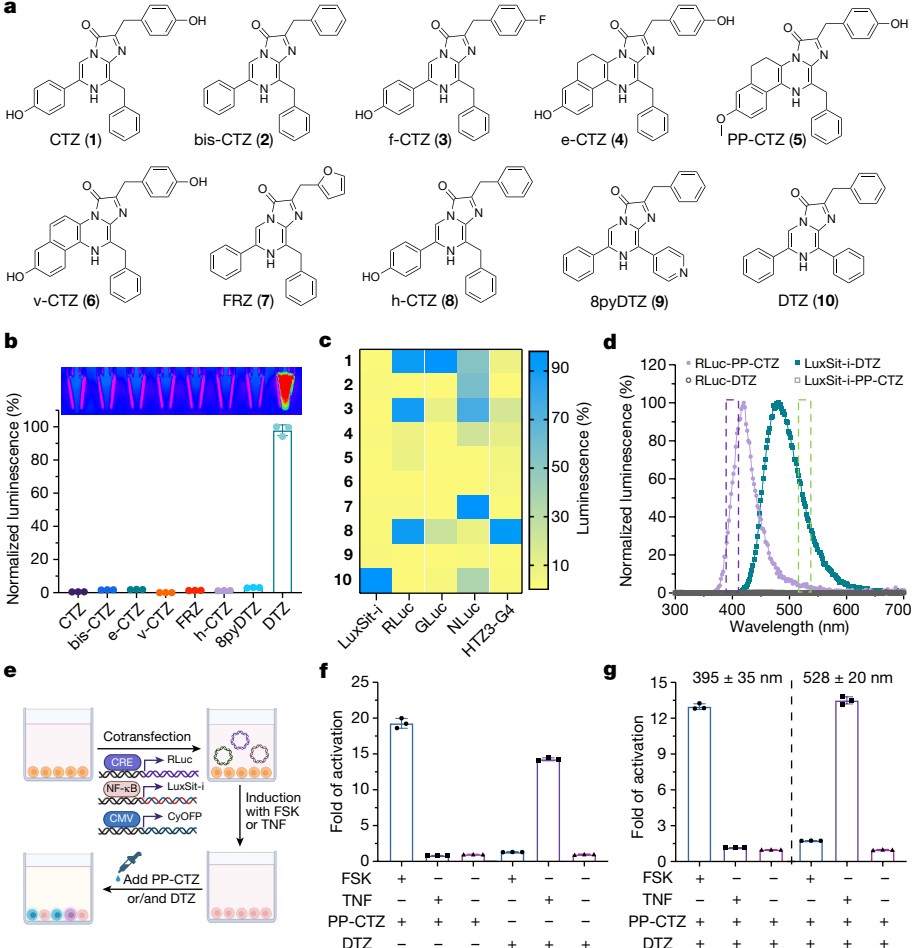

**Fig. 4 | High substrate specificity of de novo luciferases allows multiplexed bioassay. a**, Chemical structures of coelenterazine substrate analogues. **b**, Normalized activity of LuxSit-i on selected luciferin substrates. Luminescence image (top) and signal quantification (bottom) of the indicated substrate in the presence of 100 nM LuxSit-i. LuxSit-i has high specificity for the design target substrate, DTZ. **c**, Heat map visualization of the substrate specificity of LuxSit-i; *Renilla* luciferase (RLuc); *Gaussia* luciferase (GLuc); engineered NLuc from *Oplophorus* luciferase; and the de novo luciferase (HTZ3-G4) designed for h-CTZ. The heat map shows the luminescence for each enzyme on each substrate; values are normalized on a per-enzyme basis to the highest signal for that enzyme over all substrates. **d**, The luminescence emission spectrum of LuxSit-i-DTZ (green) and RLuc-PP-CTZ (purple) can be spectrally resolved by 528/20 and 390/35 filters (shown in dashed bars) and only recognize the cognate substrate. **e**, Schematic of the multiplex luciferase assay. HEK293T cells

transiently transfected with CRE-RLuc, NF-κB-LuxSit-i and CMV-CyOFP plasmids were treated with either forskolin (FSK) or human tumour necrosis factor (TNF) to induce the expression of labelled luciferases. **f,g**, Luminescence signals from cells can be measured under either substrate-resolved or spectrally resolved methods by a plate reader. **f**, For the substrate-resolved method, luminescence intensity was recorded without a filter after adding either PP-CTZ or DTZ. **g**, For the spectrally resolved method, both PP-CTZ and DTZ were added, and the signals were acquired using 528/20 and 390/35 filters simultaneously. In **f** and **g**, the bottom panel indicates the addition of FSK or TNF. Luminescence signals were acquired from the lysate of 15,000 cells in CelLytic M reagent, and the CyOFP fluorescence signal was used to normalize cell numbers and transfection efficiencies. All data were normalized to the corresponding non-stimulated control. Data are mean ± s.d. (*n* = 3).

challenge in computational enzyme design). The small size, stability and robust folding of LuxSit-i makes it well-suited in luciferase fusions to proteins of interest and as a genetic tag for capacity-limited viral vectors. On the basic science side, the small size, simplicity and high activity of LuxSit-i make it an excellent model system for computational and experimental studies of luciferase catalytic mechanism. Extending the approach used here to create similarly specific luciferases for synthetic luciferin substrates beyond DTZ and h-CTZ would considerably extend the multiplexing opportunities illustrated in Fig. 4 (particularly with the recent advances in microscopy[41]), and enable a new generation of multiplexed luminescent toolkits. More generally, our family-wide hallucination method opens up an almost unlimited number of scaffold possibilities for substrate binding and catalytic residue placement, which is particularly important when the reaction mechanism and how to promote it are not completely understood: many alternative structural and catalytic hypotheses can be readily enumerated with

shape and chemically complementary binding pockets but different catalytic residue placements. Although luciferases are unique in catalysing the emission of light, the chemical transformation of substrates into products is common to all enzymes, and the approach developed here should be readily applicable to a wide variety of chemical reactions.

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

## Reporting summary

Further information on research design is available in the Nature Portfolio Reporting Summary linked to this article.

## Data availability

Source data for Figs. 2–4 are available online. The gene sequence for LuxSit-i has been deposited to GenBank under the accession number OP820699. See Supplementary Data for design models of LuxSit and LuxSit-i. Codon-optimized plasmids encoding LuxSit-i for bacterial and mammalian expression are available through Addgene. Source data are provided with this paper.

## Code availability

The Rosetta macromolecular modelling suite (https://www.rosettacommons.org) is freely available to academic and non-commercial users. Commercial licences for the suite are available through the University of Washington Technology Transfer Office. The source code for RIF docking implementation is freely available at https://github.com/rifdock/rifdock. All relevant scripts and an accompanying Jupiter notebook for family-wide hallucination scaffold generation are available here: https://files.ipd.uw.edu/pub/luxSit/scaffold_generation.tar.gz. All generated scaffolds are available here: https://files.ipd.uw.edu/pub/luxSit/scaffolds.tar.gz. Computational design scripts for the luciferase libraries are available here: https://files.ipd.uw.edu/pub/luxSit/luciferase_designs_methods.zip.

**Acknowledgements** We thank B. Wicky and R. Kibler for assistance on CD measurements; X. Li for help with mass spectrometry analysis of proteins; D. Feldman for the initial investigation of cell imaging; L. Milles for optimizing the golden gate assembly protocol; and the National Natural Science Foundation of China (22103060) for providing computational resources that were used in quantum mechanics and MD simulations. Some figure panels were partially created with BioRender.com (Fig. 4e and Extended Data Figs. 2 and 10c). Research reported in this publication was partially supported by the National Institute of Biomedical Imaging and Bioengineering of the National Institutes of Health under award number K99EB031913. The content is solely the responsibility of the authors and does not necessarily represent the official views of the National Institutes of Health. We acknowledge funding from the Howard Hughes Medical Institute (Y.K., B.C. and D.B.); the Henrietta and Aubrey Davis Endowed Professorship in the University of Washington Department of Biochemistry (D.B.); the NIH Pathway to Independence Award (A.H.-W.Y., K99EB031913); the United World Antiviral Research Network (UWARN) (one of the Centers for Research in Emerging Infectious Diseases; CREIDs) NIAID 1 U01 AI151698-01 (D.B. and A.H.-W.Y.); the Audacious Project at the Institute for Protein Design (D.B. and A.H.-W.Y.); the Open Philanthropy Project Improving Protein Design Fund (D.B.); the Novo Nordisk Foundation (C.N., NNF18OC0030446), a Washington Research Foundation Fellowship (S.P.); E. and W. Schmidt, by recommendation of the Schmidt Futures program (D.T., G.R.L., J.Z.Z., L.C., S.H., M.D., L.C. and D.B.); and the National Science Foundation (K.N.H., D.E. and P.M., CHE-1764328, and OCI-1053575 to XSEDE).

**Author contributions** D.B. supervised this work. A.H.-W.Y. conceived the project and performed the experimental characterization. A.H.-W.Y., Y.K. and C.N. conceptualized and investigated the initial design strategies and A.H.-W.Y. performed the computational design of luciferases. C.N. conceptualized and implemented the family-wide hallucination pipeline. C.N., D.T., S.J.P. and I.A. performed the scaffold design. S.J.P. performed the ProteinMPNN sequence design of NTF2 scaffolds. K.N.H. mentored D.E. and P.M. for MD and quantum mechanics calculations and contributed to the writing of the mechanistic parts of the manuscript. G.R.L. developed the computational SSM simulation. J.Z.Z. prepared mammalian cells and performed microscopy imaging. B.C. and L.C. integrated the tuning file function into RifDock. J.D. developed ProteinMPNN. S.H. helped assemble the designed library. M.D. and L.C. assisted with protein expression and purification. A.H.-W.Y., C.N. and D.B. wrote the initial manuscript. All authors discussed the results and contributed to the final manuscript.

**Competing interests** A.H.-W.Y., C.N., Y.K., D.T., S.J.P., I.A. and D.B. are co-inventors in several provisional patent applications (application numbers 63/300171, 63/300178, 63/381922 and 63/381924 submitted by the University of Washington) covering the de novo luciferases and protein scaffolds described in this Article. A.H.-W.Y., C.N., J.Z. and D.B. are stockholders of Monod Bio, a company that aims to develop the inventions described in this manuscript. The remaining authors declare no competing interests.

**Additional information**
**Correspondence and requests for materials** should be addressed to Andy Hsien-Wei Yeh or David Baker.

**Extended Data Fig. 1 | Proposed catalytic mechanism of coelenterazine-utilizing luciferases.** Density functional theory (DFT) calculation suggested that the formation of an anionic state is the essential electron source for the activation of triplet oxygen ($^3O_2$). Supported by both theoretical[25,26] and experimental evidence[27,28], the next oxygenation process is likely through a single-electron transfer (SET) mechanism in which the surrounding reaction field could highly influence the change of Gibbs free energy ($\Delta G_{SET}$). Finally, the thermolysis of a dioxetane light emitter intermediate can produce photons via the mechanism of gradually reversible charge-transfer-induced luminescence (GRCTIL), which is generally exergonic. As all the historical pieces of evidence are based on calculations in the virtual solvents or chemiluminescence in ideal organic solvents, the detailed mechanism of a luciferase-catalysed luminescence reaction has remained unclear. We proposed that the key step of the enzyme is to promote the formation of an anionic state and create a suitable environment to facilitate efficient SET. Hence, the goal of this study is to design an enzyme reaction field surrounding the substrate to stabilize the anionic substrate state and alter the local proton activity, solvent polarity, and hydrophobicity for the efficient activation of $^3O_2$.

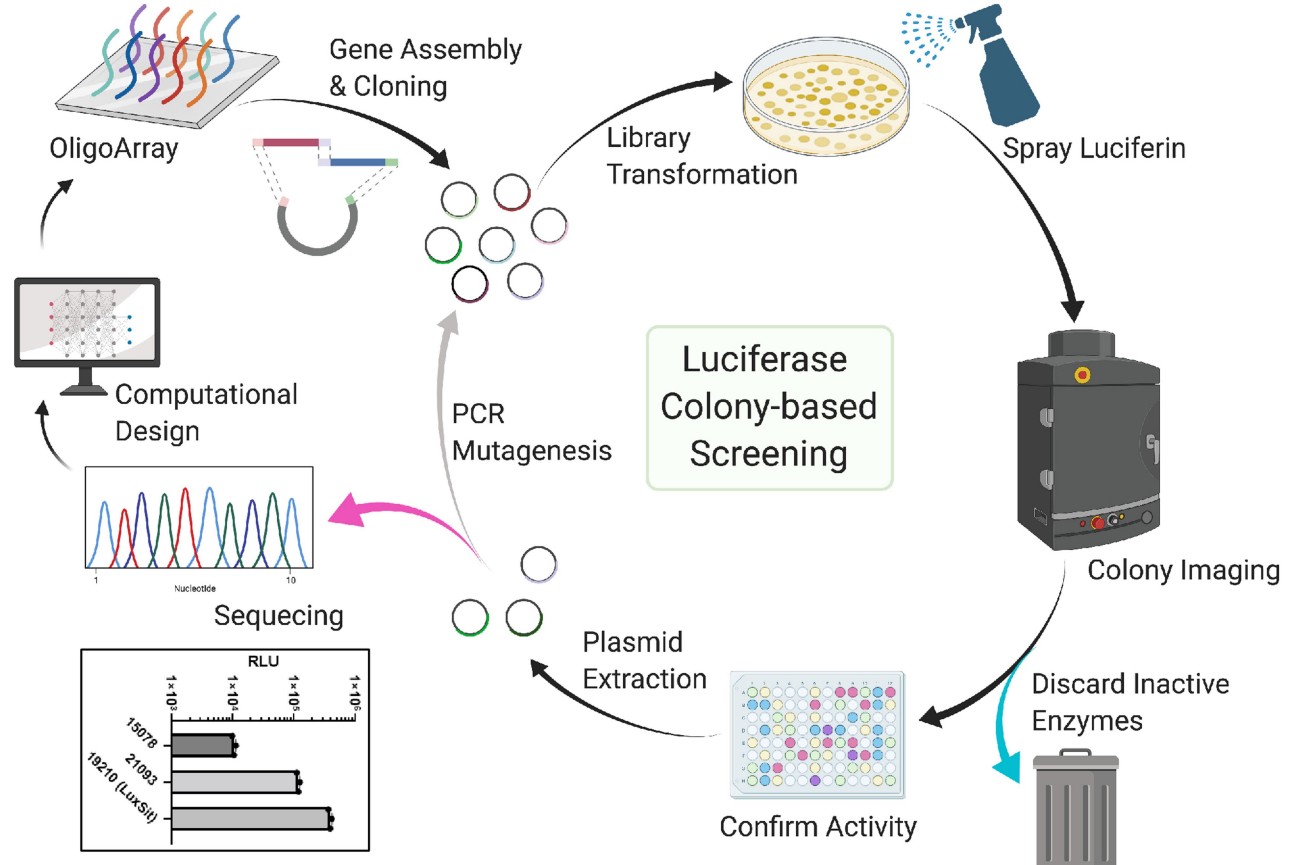

**Extended Data Fig. 2 | Schematic representation of colony-based luciferase screening.** Computationally designed DNA sequences were purchased in an oligo array, where the fragments were amplified by PCR, assembled, and ligated into a pBAD bacterial expression vector. The plasmid library was used to transform DH10B cells. Each colony grown on the LB agar plate represented one luciferase design. The plates were sprayed with DTZ solution and imaged to identify active colonies using a ChemiDoc imager. Selected colonies were inoculated in 96-well plates, expressed, and purified to confirm individual luciferase activity. Plasmids can then be individually sequenced to point out active design models that provide insights into the design principle and enzyme functions or can be subjected to random mutagenesis for further evolution. Insert: three luciferases were identified from this screening. We refer to the most active and DTZ-specific luciferase as "LuxSit".

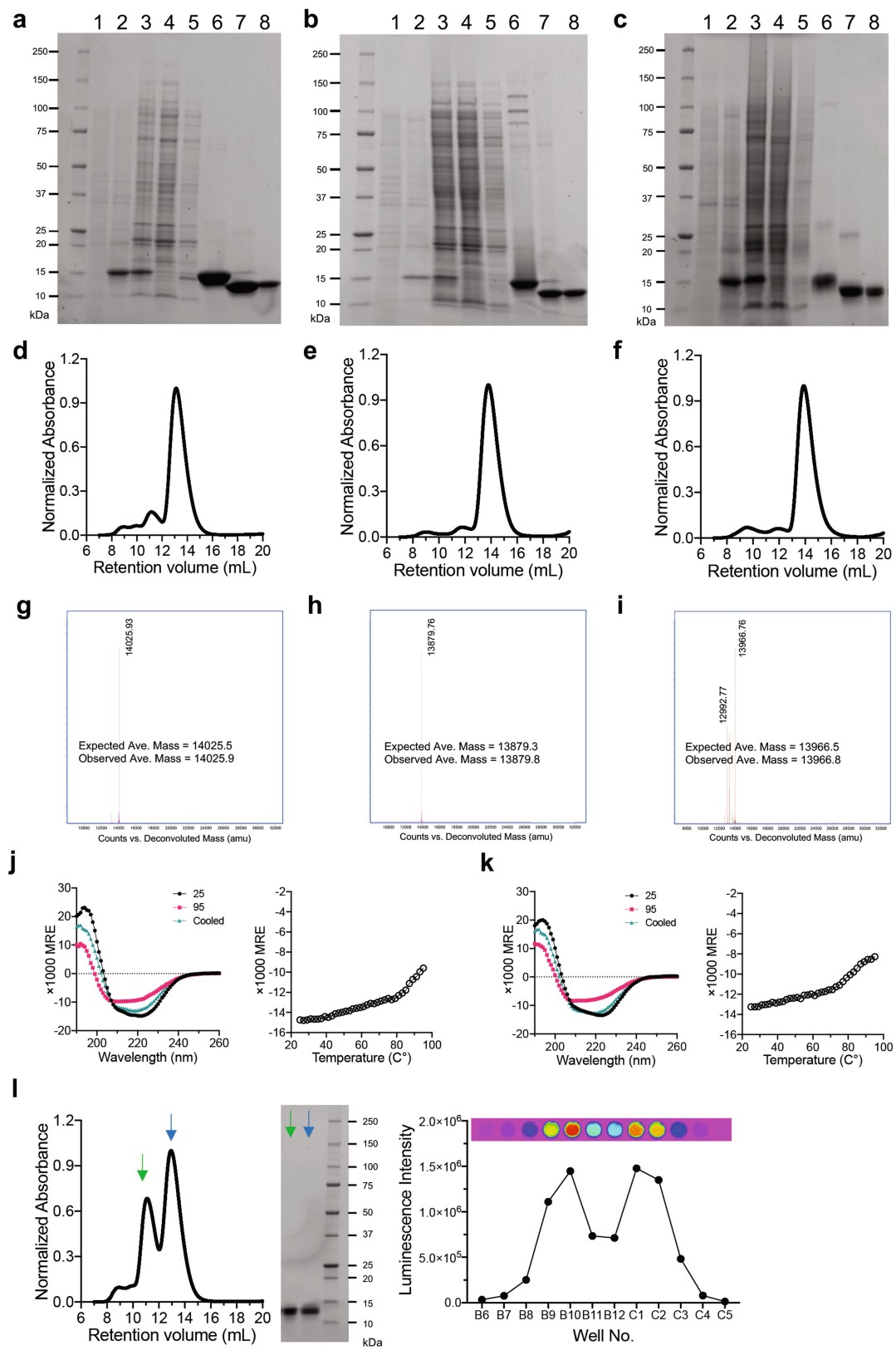

**Extended Data Fig. 3 | Expression, purification and structural characterization of LuxSit variants. a–c**, The recombinant expression of **a**, LuxSit, **b**, LuxSit-i, and **c**, LuxSit-f in *E. coli*. Annotations for each lane are the following – 1: Pre-IPTG; 2: Post-IPTG; 3: Soluble lysate; 4: Flow-through; 5: Wash; 6: Elusion; 7: Post-TEV cleavage; 8: Post-SEC. **d–f**, Size-exclusion chromatography of the purified **d**, LuxSit; **e**, LuxSit-i; and **f**, LuxSit-f monomer. **g–i**, Deconvoluted mass spectrum of **g**, LuxSit, **h**, LuxSit-i, and **i**, LuxSit-f. **j,k**, Far-ultraviolet circular

dichroism (CD) spectra (Left panel) of **j**, LuxSit-i; and **k**, LuxSit-f at 25 °C (black line), 95 °C (red line) and cooled back to 25 °C (green line). CD melting curve at 220 nm (Right panel). **l**, Dimeric SEC peak was observed when LuxSit-i was concentrated to a high concentration (~50 μM) in Tris pH 8.0 buffer. Both dimeric and monomeric SEC fractions showed the expected size on SDS–PAGE and both peaks were catalytically active to emit luminescence in the presence of 25 μM DTZ.

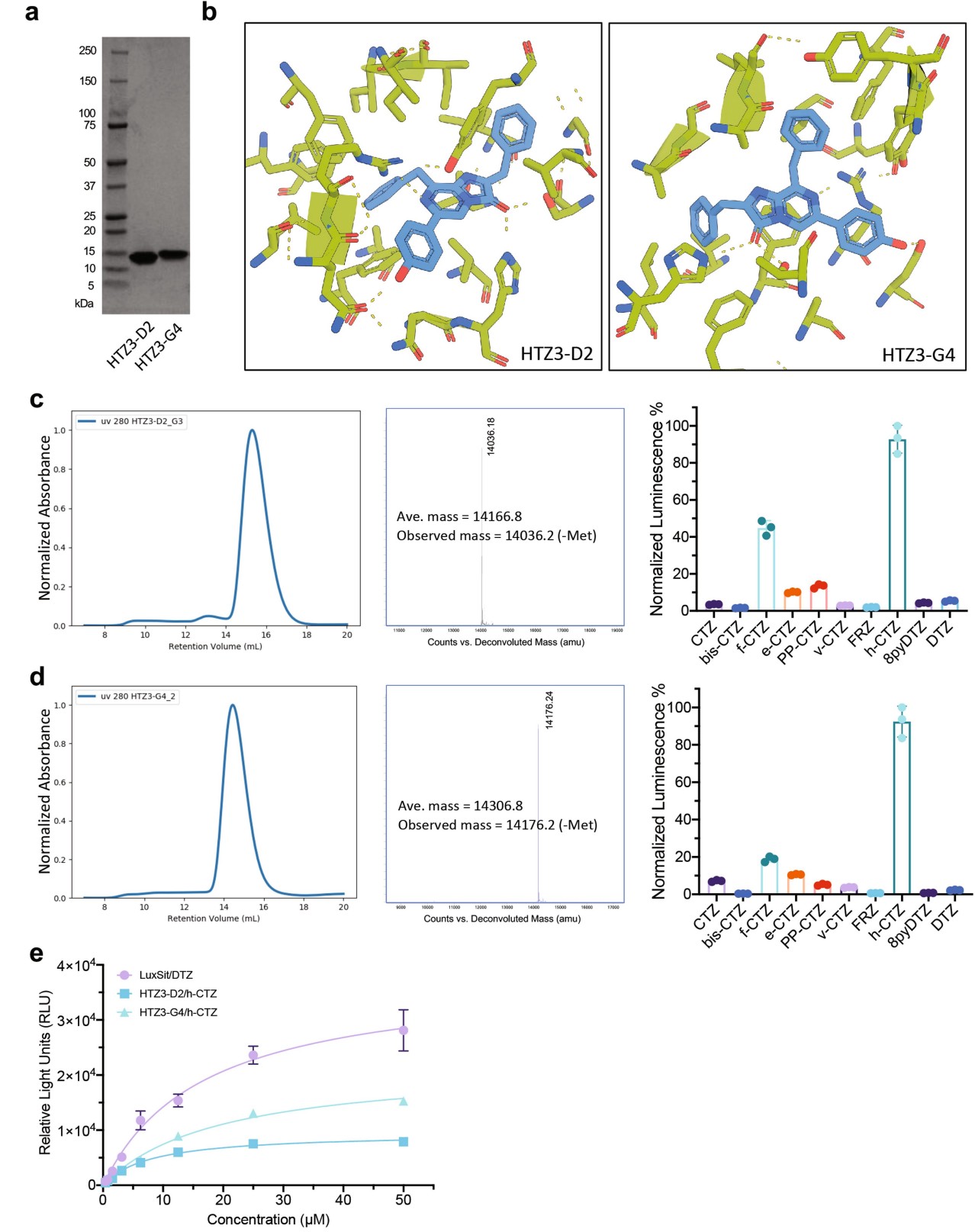

**Extended Data Fig. 4 | Expression, purification and activity measurement of selected de-novo-designed luciferases for h-CTZ. a**, Coomassie-stained SDS–PAGE of HTZ3-D2 and HTZ3-G4 purified from recombinant expression in *E. coli*. **b**, Magnified views of HTZ3-D2 (left panel) and HTZ3-G4 (right panel) illustrated the side-chain preorganization of luciferase-h-CTZ interactions. **c,d**, Size-exclusion chromatography (left), deconvoluted mass spectrum (middle), and the normalized luciferase activities on selected compounds (right) of **c**, HTZ3-D2 and **d**, HTZ3-G4, which suggested high specificity for the design target substrate, h-CTZ. **e**, Substrate concentration dependence of LuxSit (w/ DTZ), HTZ3-D2 (w/ h-CTZ), and HTZ3-G4 (w/ h-CTZ) activity in PBS. All data points were fitted to the Michaelis-Menten equation. HTZ3-D2 and HTZ3-G4 showed $K_m$ values of 7.9 and 19.5 μM with ~25% and ~58% $I_{max}$ of LuxSit, respectively. Data are presented as mean ± s.d. (*n* = 3).

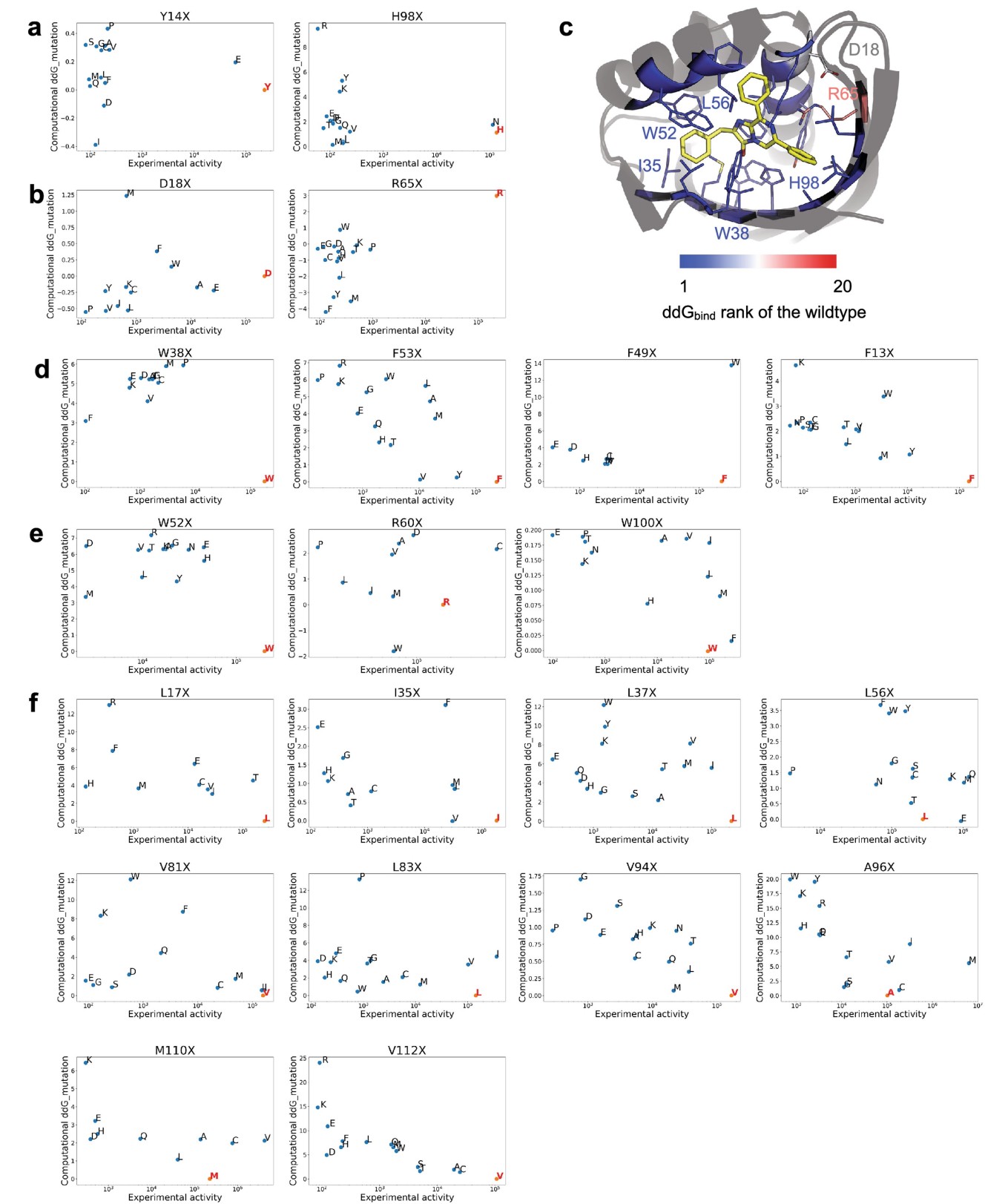

**Extended Data Fig. 5 | Predicted changes in substrate-binding free energy from binding-site mutations.** The calculated ddG$_{bind}$ of each mutation was plotted as a function of the relative average experimental luciferase activity. The ddG$_{bind}$ of hypothetical catalytic residues: **a**, Tyr14–His98 and **b**, Asp18–Arg65 dyads were generally not the lowest, which suggested that these designed catalytic residues are not favourable for substrate binding. Red dots represent the wild-type (LuxSit) amino acids. The rank of wild-type ddG$_{bind}$ for each position screened for activity is shown with a heat map in **c**. **d**–**f**, The wild-type ddG$_{bind}$ of the residues designed for **d**,**e**, π–π stacking or **f**, hydrophobic interactions were the lowest compared to the mutation ddG$_{bind}$ values. This shows that the sequence is near-optimal for substrate binding and the design model is reliable.

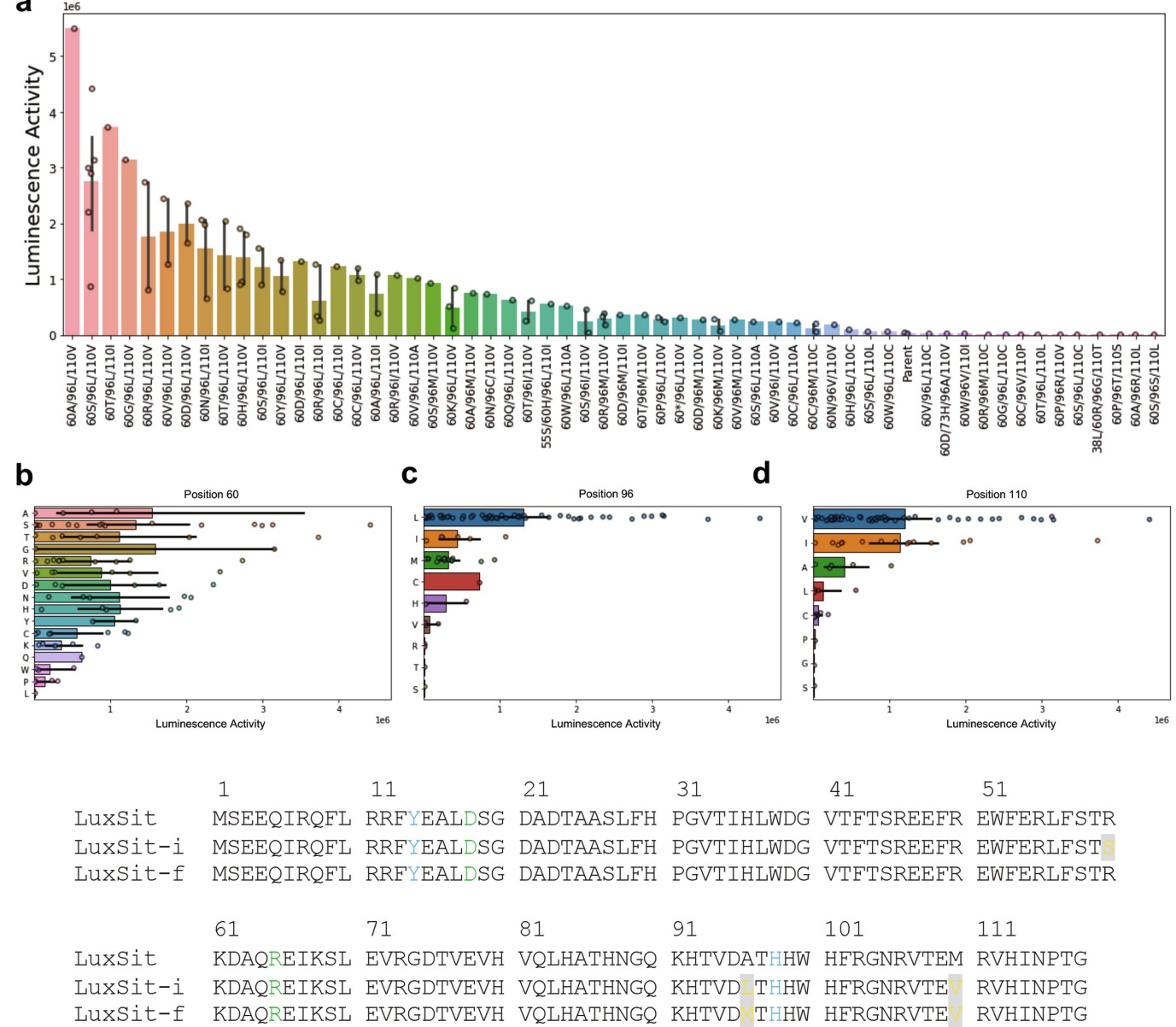

**Extended Data Fig. 6 | Screening of a randomized NNK library at 60, 96 and 110 positions and sequence alignment between LuxSit and its variants.** We generated a fully randomized library at 60, 96, and 110 positions to screen all possible combinations exhaustively. After the colony-based screening, we identified many colonies with strong luciferase activities with DTZ. Each colony was expressed individually in each well of 96-well plates (1 mL culture) and purified accordingly (see Supplementary Methods). **a**, Individual luminescence activity of each selected mutant was plotted and compared to the parent, LuxSit. Luminescence activities were measured in the presence of 25 μM DTZ. Luminescence activity (RLU) was shown as the integrated signal over the first 15 min. Statistical analysis of the amino acid frequency versus the

luciferase activity at residue **b**, 60, **c**, 96, and **d**, 110. Data are presented as mean ± s.d. (n varies across each bar as the mutants were selected from a randomized library). Arg60 is confirmed to be mutable among all selected mutants as Arg60 may be structurally less well-defined because it emanates from a loop and has no hydrogen-bonding partner. Ala96 prefers larger side-chain substitutions (Leu, Ile, Met, and Cys), and Met110 favours hydrophobic residues (Val, Ile, and Ala). A newly discovered variant (R60S/A96L/M110V) with more than 100-fold higher photon flux over LuxSit was assigned LuxSit-i for its high brightness. In the sequence alignment, mutations are highlighted in yellow fonts and grey backgrounds. The conserved catalytic dyads of Asp18–Arg65 and Tyr14–His98 are in green and blue fonts.

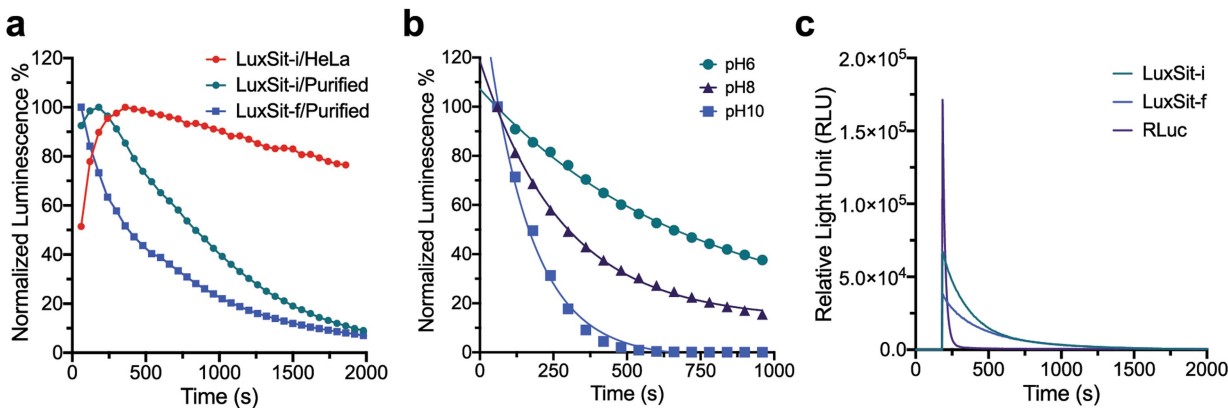

**Extended Data Fig. 7 | Additional characterization of LuxSit variants.**
 **a**, Normalized emission kinetics of 15,000 intact HeLa cells expressing LuxSit-i (red), 100 nM purified LuxSit-i (green), or 100 nM purified LuxSit-f (blue) in the presence of 50 µM DTZ. The more extended emission kinetics in HeLa cells is likely due to the diffusion rate of DTZ across cell membranes. **b**, Normalized luminescence decay curves of LuxSit-i in various pH buffers revealed a pH-dependent catalytic mechanism. **c**, Luminescent quantum yield was estimated from the integrated luminescence signal until completely converting 125 pmol substrates to photons in the presence of 50 nM corresponding luciferase (see Supplementary Methods). Data are presented as mean ($n$ = 3).

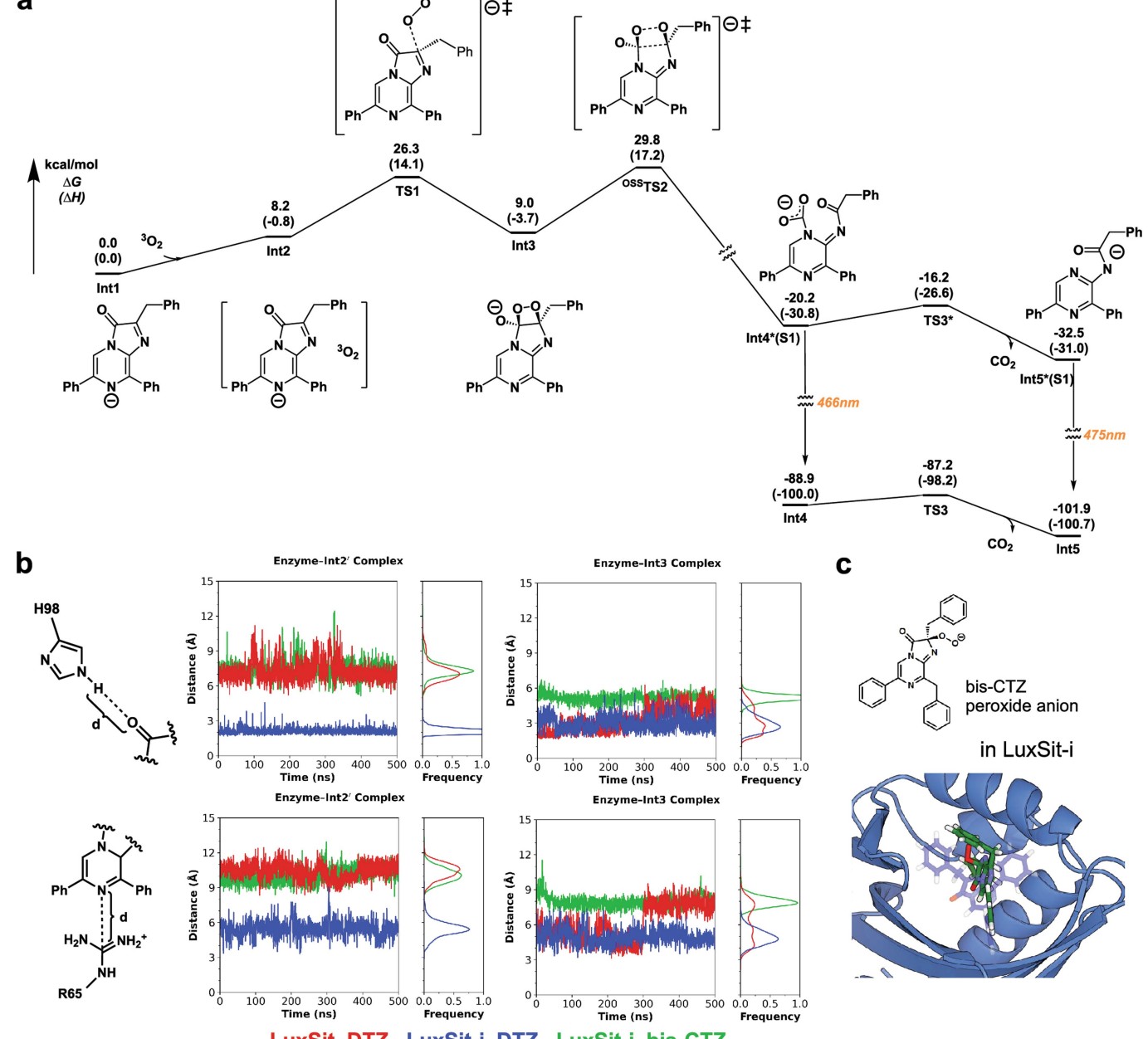

**Extended Data Fig. 8 | Free-energy profile of DTZ chemiluminescence and MD simulations of proposed protein–intermediate complexes. a**, The free-energy profile calculated by density functional theory (DFT) shows triplet oxygen can react directly with the anionic species of DTZ (**Int1**) through the reactant complex **Int2** and **TS1**. The dioxetane intermediate **Int3** then cleaves in an open shell singlet transition state $^{oss}$**TS2** to form excited intermediate **Int4\***, which rapidly extrudes $CO_2$ and forms the emissive product **Int5**. Note: either **Int4\*** or **Int5\*** emit in the observed region, but the lifetime of **Int4\*** is very short and likely completely converts to **Int5\*** before emission. **b**, **Int2** and **Int3** were docked into both LuxSit and LuxSit-i and the bindings were evaluated by molecular dynamics (MD). The distances between His98 to O1 (top row) and Arg65 to N1 (bottom row) of the substrate were plotted throughout 500 ns MD

simulations. LuxSit-i (blue trace) binds **Int2′** (middle) considerably better than LuxSit does (red trace), suggesting that the mutations of LuxSit-i provide a binding pocket more complimentary to **TS1**. This binding orientation brings N1 of the substrate much closer to Arg65, providing better charge stabilization for the high energy transition state. **c**, Docking of the peroxide anion form of bis-CTZ into the pocket of LuxSit-i; blue overlay represents DTZ in the original design model. During MD simulation, the added benzylic carbon of bis-CTZ (green trace) disrupts the shape complementarity between LuxSit-i and the transition states (**TS1** and **TS2**), reducing the charge stabilization by Arg65. This charge stabilization is necessary for the reaction to proceed, explaining the high substrate specificity of LuxSit-i for DTZ over bis-CTZ.

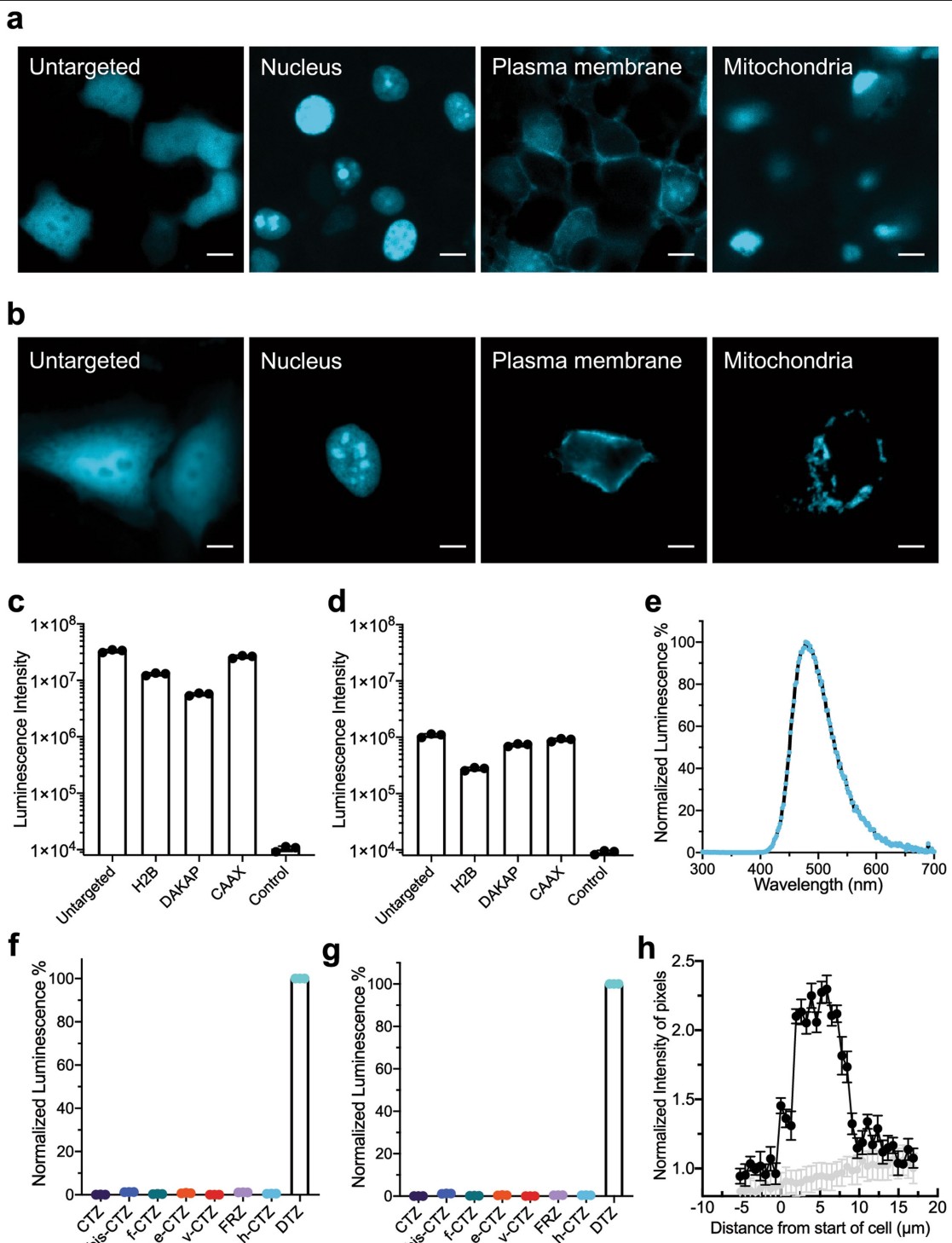

**Extended Data Fig. 9 | Expression, localization and luminescence activity of LuxSit-i in live HEK293T and HeLa cells. a**,**b**, Fluorescence imaging of live **a**, HEK293T and **b**, HeLa cells expressing LuxSit-i-mTagBFP2, which is untargeted or localized to the nucleus (Histone2B), plasma membrane (KRasCAAX), or mitochondria (DAKAP) cellular compartments. Scale bar: 10 µm. **c**,**d**, Luminescence signals were measured with 15,000 intact **c**, HEK293T or **d**, HeLa cells in the presence of 25 µM DTZ in DPBS. Transfection efficiencies range from 60-70% for HEK293T cells and 5-10% for HeLa cells. **e**, Luminescence emission spectra acquired from LuxSit-i expressing HEK293T cells is consistent with the emission spectra of recombinant LuxSit-i purified from *E. coli*. **f**,**g**, Luminescence signals were measured with 15,000 **f**, intact LuxSit-i expressing HEK293T cells or **g**, cell lysate in the presence of 25 µM indicated substrate in DPBS. Luminescence intensities were normalized to DTZ signal, showing high DTZ specificity over other substrates in cell-based assays. Data were shown as total luminescence signal over the first 20 min ± s.d. (*n* = 3). **h**, Normalized luminescence intensity profile of lines traversing across different cells (*n* = 10) of main Fig. 3c luminescence image; grey lines represent untransfected cells. Error bars represent ± SEM.

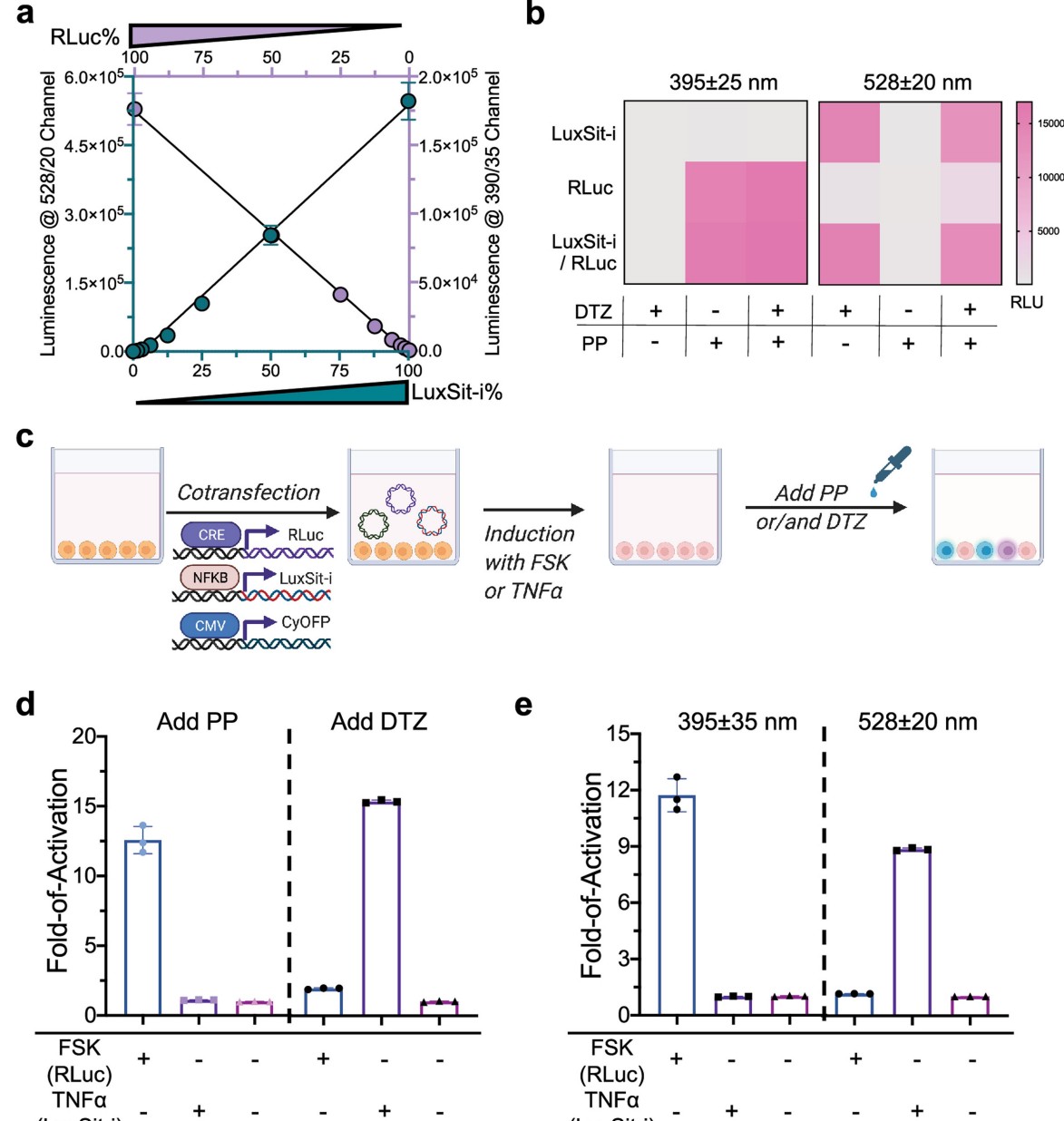

**Extended Data Fig. 10 | Substrate specificity of LuxSit-i and spectrally resolved luciferase–luciferin pairs allow multiplexed bioassay. a**, The orthogonality relationship between LuxSit-i-DTZ and RLuc-PP-CTZ (Prolume Purple, methoxy e-Coelenterazine) luminescent pairs. Indicated percentages of each luciferase were mixed at different ratios totalling 100%. After the addition of both 25 μM DTZ and PP-CTZ substrates, filtered light from 528/20 and 390/35 channels were measured simultaneously. **b**, Heat map shows the luminescence signal for individual luciferase (100 nM) or 1:1 mixture in the presence of the cognate or non-cognate (DTZ or PP-CTZ or both) substrates. Response signals were acquired by a Neo2 plate reader with 528/20 and

390/35 nm filters simultaneously. **c**, Multiplex luciferase assay in live HEK293T after co-transfection of CRE-RLuc, NFκB-LuxSit-i, and CMV-CyOFP plasmids and stimulation by Forskolin (FSK) or human TNF. **d,e**, 15,000 intact cells were assayed (see Supplementary Methods) by either **d**, substrate-resolved or **e**, spectrally resolved modes after adding DTZ, PP-CTZ, or both DTZ and PP-CTZ in DPBS without cell lysis. Area scanning of the CyOFP fluorescence signal was used to estimate cell numbers and transfection efficiency. The reported unit was RLU/a.u.; relative light units/fluorescence intensity measurements at Ex./Em. = 480/580 nm. All data were normalized to the corresponding non-stimulated control. Data are presented as mean ± s.d. (*n* = 3).

# Reporting Summary

## Statistics

For all statistical analyses, confirm that the following items are present in the figure legend, table legend, main text, or Methods section.

| n/a | Confirmed | |
|---|---|---|
| ☐ | ☒ | The exact sample size (*n*) for each experimental group/condition, given as a discrete number and unit of measurement |
| ☐ | ☒ | A statement on whether measurements were taken from distinct samples or whether the same sample was measured repeatedly |
| ☒ | ☐ | The statistical test(s) used AND whether they are one- or two-sided *Only common tests should be described solely by name; describe more complex techniques in the Methods section.* |
| ☒ | ☐ | A description of all covariates tested |
| ☒ | ☐ | A description of any assumptions or corrections, such as tests of normality and adjustment for multiple comparisons |
| ☐ | ☒ | A full description of the statistical parameters including central tendency (e.g. means) or other basic estimates (e.g. regression coefficient) AND variation (e.g. standard deviation) or associated estimates of uncertainty (e.g. confidence intervals) |
| ☒ | ☐ | For null hypothesis testing, the test statistic (e.g. *F*, *t*, *r*) with confidence intervals, effect sizes, degrees of freedom and *P* value noted *Give P values as exact values whenever suitable.* |
| ☒ | ☐ | For Bayesian analysis, information on the choice of priors and Markov chain Monte Carlo settings |
| ☒ | ☐ | For hierarchical and complex designs, identification of the appropriate level for tests and full reporting of outcomes |
| ☒ | ☐ | Estimates of effect sizes (e.g. Cohen's *d*, Pearson's *r*), indicating how they were calculated |

*Our web collection on statistics for biologists contains articles on many of the points above.*

## Software and code

Policy information about availability of computer code

| Data collection | The Rosetta macromolecular modelling suite (https://www.rosettacommons.org) is freely available to academic and non-commercial users. Commercial licences for the suite are available through the University of Washington Technology Transfer Office.<br><br>The source code for RIF docking implementation is freely available at https://github.com/rifdock/rifdock.<br><br>All relevant scripts and an accompanying Jupiter notebook for family-wide hallucination scaffold generation are available here: https://files.ipd.uw.edu/pub/luxSit/scaffold_generation.tar.gz<br><br>All generated scaffolds are available here: https://files.ipd.uw.edu/pub/luxSit/scaffolds.tar.gz<br><br>Design scripts for active LuxSit are available here: https://files.ipd.uw.edu/pub/luxSit/luciferase_designs_methods.zip |
|---|---|
| Data analysis | The molecular mass of each protein was deconvoluted by Bioconfirm software (10.0) using a total entropy algorithm. Microscopic fluorescence and luminescence images were analyzed using NIS Elements 5.30 software. Data were analyzed and plotted using GraphPad Prism 8, Python 3.8, DNAWorks2.0, seaborn0.12.1, and matplotlib3.6.2. |

For manuscripts utilizing custom algorithms or software that are central to the research but not yet described in published literature, software must be made available to editors and reviewers. We strongly encourage code deposition in a community repository (e.g. GitHub). See the Nature Portfolio guidelines for submitting code & software for further information.

## Data

Policy information about availability of data

All manuscripts must include a data availability statement. This statement should provide the following information, where applicable:
- Accession codes, unique identifiers, or web links for publicly available datasets
- A description of any restrictions on data availability
- For clinical datasets or third party data, please ensure that the statement adheres to our policy

Source data for Fig. 2, 3, and 4 are available online. The gene sequence for LuxSit-i has been deposited to GenBank under the accession number OP820699. Design models of LuxSit and LuxSit-i are available online in LuxSit_models.zip. Codon-optimized plasmids encoding LuxSit-i for bacterial and mammalian expression are available through Addgene.

## Human research participants

Policy information about studies involving human research participants and Sex and Gender in Research.

| | |
|---|---|
| Reporting on sex and gender | N/A |
| Population characteristics | N/A |
| Recruitment | N/A |
| Ethics oversight | N/A |

Note that full information on the approval of the study protocol must also be provided in the manuscript.

# Field-specific reporting

Please select the one below that is the best fit for your research. If you are not sure, read the appropriate sections before making your selection.

☒ Life sciences          ☐ Behavioural & social sciences          ☐ Ecological, evolutionary & environmental sciences

For a reference copy of the document with all sections, see nature.com/documents/nr-reporting-summary-flat.pdf

# Life sciences study design

All studies must disclose on these points even when the disclosure is negative.

| | |
|---|---|
| Sample size | No statistical methods were used to pre-determine the sample size. The number of designs we ordered is limited by the availability of adaptor pairs and the size of Oligo pool. |
| Data exclusions | No sample was excluded from the data analysis. |
| Replication | Results were reproduced using different batches of pure proteins on different days. Coomassie-stained SDS PAGE gels were done at least twice for each experiment. Microscopic fluorescence and luminescence images were repeated in biological triplicate with similar results. |
| Randomization | For the cellular imaging and assay, cells were randomly separated into experimental groups. |
| Blinding | Researchers were not blinded to the experiments described in this study. |

# Reporting for specific materials, systems and methods

We require information from authors about some types of materials, experimental systems and methods used in many studies. Here, indicate whether each material, system or method listed is relevant to your study. If you are not sure if a list item applies to your research, read the appropriate section before selecting a response.

## Materials & experimental systems

| n/a | Involved in the study |
|-----|----------------------|
| ☒ | ☐ Antibodies |
| ☐ | ☒ Eukaryotic cell lines |
| ☒ | ☐ Palaeontology and archaeology |
| ☒ | ☐ Animals and other organisms |
| ☒ | ☐ Clinical data |
| ☒ | ☐ Dual use research of concern |

## Methods

| n/a | Involved in the study |
|-----|----------------------|
| ☒ | ☐ ChIP-seq |
| ☒ | ☐ Flow cytometry |
| ☒ | ☐ MRI-based neuroimaging |

# Eukaryotic cell lines

Policy information about cell lines and Sex and Gender in Research

| | |
|---|---|
| Cell line source(s) | HEK293T (ATCC, CRL-11268) and HeLa (ATCC, CCL-2) |
| Authentication | Autheticated by vendors. Cells were not authenticated in the lab |
| Mycoplasma contamination | All cells were tested negative for Mycoplasma by the provider, and it was not further confirmed in the lab. |
| Commonly misidentified lines (See ICLAC register) | no commonly misidentified cell lines were used in the study |

