## [Peer Review File · Nature]

Manuscript Title: De novo design of luciferases using deep learning

Reviewer Comments & Author Rebuttals

Reviewer Reports on the Initial Version:

Referees' comments:

Referee #1:

Over the last 20 years there has been exciting progress in the field of computational protein design as methods have been developed for creating proteins that adopt predetermined structures or complexes, but the de novo design of highly active and useful enzymes has remained largely an unsolved problem. This paper describes a multi-step process for designing de novo enzymes and uses the approach to create a novel luciferase that should immediately be useful to biologists. This is an exciting result, and the methods are likely to be applicable for the design of other enzymes and biosensors. A key step in the design process was the use of new methods in deep learning to generate a large set of protein scaffolds with a common fold but alternative placement of amino acid side chains around a putative active site. It is likely that this is just the first of many studies that will make use of deep learning methods to help in the creation of functional proteins.

Specific comments:

- A large set of designs (>7000) were screened to identify the protein, LuxSit, characterized in this study. Were any of the designs besides LuxSit active? How did their activity compare to LuxSit?
- Site-saturation mutagenesis identified two mutations, A96M and M110V, that increased the activity of the initial design by over 100-fold. Are these mutations compatible with the substrate docked in the conformation used for the initial sequence design. If not, how much adjustment of the substrate positioning is needed to make room for the mutations. Does modeling of the mutations provide any hypotheses for how they are improving activity?
- From the text in the main body of the manuscript it is hard to decipher how the RifGen and RifDock methods work. Can this section of the manuscript be made more understandable to a general molecular and structural biology audience? On average, how many side chains were placed in the scaffold by RifGen/RifDock. In LuxSit, how many of the active site residues came from RifDock?
- All of the non-active substrates tested in Fig. 4 have an extra benzylic carbon when compared to DTZ. Based on your model of the active site, is it possible to rationalize why this extra carbon would impede activity? It would be interesting to test analogs of DTZ that do not have this additional carbon but have changes at other positions in the molecule.

Referee #2:

This manuscript describes a new, partly computational approach, based on a combination of deep learning and more 'classical' physics-based optimization, to design enzymes with optimized substrate selectivities. The authors apply the approach to NTF2 proteins to design an optimized

luciferase.

I was asked to focus on the deep learning component of the manuscript and so I will restrict my comments accordingly. On the whole, the approach described here does not deviate substantially from the trRosetta-based hallucination approach that was previously published (<https://www.nature.com/articles/s41586-021-04184-w>), including the use of KL divergence and MCMC search. The main innovation is a new fold-specific loss function that incentivizes the model to focus on NTF2-like topologies, by penalizing it when its hallucinations deviate from the fold. This is an interesting idea and was applied effectively in this work. On its own the novelty presented by this change is somewhat limited, but of course this is only one component of the overall manuscript.

My main specific comments center around the description of the method, specifically section 3.2 in Methods, which is rather sparse and can use substantial beefing up. In particular:

- The sentence "For each seed structure, we used a structure-based sequence alignment 35 method (see 3.3) to find equivalent positions between that and a set of crystal structures (χ) trimmed to only include "ideal," designable elements." refers to ideal and designable elements. How are these formally defined? Right now the description is entirely qualitative and it's unclear how these elements are actually selected.

- The sentence "For pairs of positions with at least 5 crystal structure equivalent positions, smooth probability distributions were computed by applying a Gaussian blur to distances, a von Mises blur along omega angle, and von Mises-Fisher blur along phi and theta angles." gives high-level detail to how the loss function is constructed but no numerical details, making this irreproducible. The actual functional form, including hyperparameter choices, should be given.

- This sentence, "Finally, we converted this to an energy by taking the negative log-likelihood 40 and calculated the expected energy given the trRosetta probability distribution to encourage the network to seek out the consensus structure, while still allowing deviations where needed." comes across as very vague. I really don't understand in detail what is happening here. Equations and/or pseudocode would communicate this much more easily. Right now the exposition feels like a high-level overview of what was done with few details on the actual implementation.

- In terms of the four different kinds of MCMC moves, are they all equally likely?

Finally, a code release would ameliorate the ambiguities described above by actually providing the tools to reproduce the computational experiments. Given the excellent standards set by the DeepMind team with AlphaFold2 and their code release, I strongly advocate that a similar code release be made here, with the full protocols needed, including Rosetta-based ones, to go from the initial sequence search for NTF2 homologs to the final designed structures.

Referee #3:

Baker and colleagues report the de novo design of a new luciferase enzyme. Computational approaches were used to identify candidate protein structures that could accept a "ready-to-react" luciferin analog and use it to generate light. A set of designer sequences was screened and a light emissive structure was identified. Further mutagenesis and biochemical assays were performed. The utility of the new luciferase was also evaluated in cultured cell models and in conjunction with other luciferin analogs. Overall, this work advances the field of protein design, making inroads into

the design of enzyme catalysts—a formidable challenge. The experiments were well designed and executed. I am a little underwhelmed by current manuscript, though, based on the data provided and lack of discussion in key areas. My comments and concerns are outlined below.

1. The manuscript is a little light on discussion regarding key design principles. When the NTF2-like superfamily is first mentioned (top of p. 2), the choice seems rather arbitrary. Several “pockets of appropriate size and shape” could exist. Perhaps the data in Fig. 1e-g are meant to provide additional rationale, but they are not discussed until later in the text. I found this section of the manuscript hard to navigate. What were the benchmarks for success? Were other protein scaffolds considered and/or attempted in this approach? Scaffolds that “didn’t work” might also be interesting from a mechanistic standpoint. Is there a way to compare the LuxSit sequence back to the original computational data set, in terms of relative ranking?

2. Related to the above point, some additional details on the addition of a key Arg residue would be helpful. Is this residue also present in native luciferase structures (especially those that have been crystallized)? It would also seem that if the luciferin anion is stabilized too much, turnover would be inhibited. Did this issue factor into the design? Did the authors consider designing binders for a different luciferin intermediates along the reaction coordinate? Stabilized dioxetanes, etc. have been used in screens by others developing artificial luciferase-like enzymes and ribozymes.

3. The manuscript hints at the generality of the approach, but this feature wasn’t quite established. Luciferase appears to be a great initial choice, given that the only other requirement beyond the luciferin is molecular oxygen.

4. While appropriate biochemical assays were performed, the mechanism of light emission is still speculative. It’s unfortunate that a crystal structure wasn’t obtained to provide additional evidence for luciferin binding, etc. Perhaps it is possible to tease out details on the importance of the anion, though, using a synthetic control compound?

5. LuxSit appears to be a viable probe for imaging studies. It would be good to see the raw light outputs alongside RLuc, GLuc, and NLuc, though (versus just normalized intensities in some cases). The light output might be too weak to pair alongside some of the brighter probes in the field for simultaneous imaging. It would also be helpful to see the raw photon data for the analog comparisons. It’s difficult to get a sense for the selectivity as plotted.

Author Rebuttals to Initial Comments:

Referee #1:

Over the last 20 years there has been exciting progress in the field of computational protein design as methods have been developed for creating proteins that adopt predetermined structures or complexes, but the de novo design of highly active and useful enzymes has remained largely an unsolved problem. This paper describes a multi-step process for designing de novo enzymes and uses the approach to create a novel luciferase that should immediately be useful to biologists. This is an exciting result, and the methods are likely to be applicable for the design of other enzymes and biosensors. A key step in the design process was the use of new methods in deep learning to generate a large set of protein scaffolds with a common fold but alternative placement of amino acid side chains around a putative active site. It is likely that this is just the first of many studies that will make use of deep learning methods to help in the creation of functional proteins.

We thank Reviewer #1 for the comments.

Specific comments:

1. A large set of designs (>7000) were screened to identify the protein, LuxSit, characterized in this study. Were any of the designs besides LuxSit active? How did their activity compare to LuxSit?

We identified three hits from the initial library containing 7982 de novo luciferases designed for DTZ. LuxSit is one to two orders of magnitude higher in activity than the other candidates. We've added the activity bar plot as an insert in the Extended Data Fig. 2.

2. Site-saturation mutagenesis identified two mutations, A96M and M110V, that increased the activity of the initial design by over 100-fold. Are these mutations compatible with the substrate docked in the conformation used for the initial sequence design. If not, how much adjustment of the substrate positioning is needed to make room for the mutations. Does modeling of the mutations provide any hypotheses for how they are improving activity?

The superposition of LuxSit and LixSit-i (R60S/A96L/M110V) models after Rosetta FastRelax suggested very subtle changes (Full-atom RMSD = 1.2 Å) in terms of substrate position and side-chain placement. We added the following description in the conclusion to emphasize this: “*This is a notable advance for computational enzyme design, as tens of rounds of directed evolution were required to obtain catalytic proficiencies in this range for a designed retroaldolase, and the structure was remodeled considerably³⁸; in contrast, the predicted differences in ligand-sidechain interactions between LuxSit and LuxSit-i are very subtle (Fig. S1b).*”.

To further investigate the effects of these mutations, we performed molecular dynamics (MD) simulations for LuxSit and LixSit-i. The result suggested that the mutations in LuxSit-i improve activity by providing a binding pocket more complementary to the forming peroxide anion transition state (**Extended Data Fig. 8b**). The primary role of the designed Arg65 is to stabilize the charge on the high-energy transition states.

3. From the text in the main body of the manuscript it is hard to decipher how the RifGen and RifDock methods work. Can this section of the manuscript be made more understandable to a general molecular and structural biology audience? On average, how many side chains were placed in the scaffold by RifGen/RifDock. In LuxSit, how many of the active site residues came from RifDock?

We revised the main text to better navigate the RIFgen/RIFdock methods (the original methodology was described in Ref. 31 and more development in Ref. 32). We also added some detailed descriptions to the Methods.

We’ve added a supplemental figure (**Fig. S1a**) showing a histogram of the number of sidechains placed on each scaffold by RIFdock (see below) and added the following description in the main text: “*An average of eight sidechain rotamers including an arginine to stabilize the anionic imidazopyrazinone core were positioned in each pocket (Fig. 1d)*”. For the ligand interactions of LuxSit, ten sidechains were initially placed by RIFdock, and five of them were mutated during RosettaDesign. Both Tyr-His and Asp-Arg dyads were from the pre-defined hydrogen bond networks.

4. All of the non-active substrates tested in Fig. 4 have an extra benzylic carbon when compared to DTZ. Based on your model of the active site, is it possible to rationalize why this extra carbon would impede activity? It would be interesting to test analogs of DTZ that do not have this additional carbon but have changes at other positions in the molecule.

Our rationale is that the shape complementary pocket was tailored for DTZ, so we expected to see high substrate selectivity. We have provided additional evidence by MD simulations, suggesting that LuxSit-i provided a shape complementary pocket to the transition state charge stabilization of DTZ but not bis-CTZ (**Extended Data Fig. 8bc**). The presence of the

benzyl ring at R₈ moves the bis-CTZ peroxide anion into an orientation where the transition state charge stabilization is no longer possible. This charge stabilization is necessary for the reaction to proceed, explaining the high substrate specificity for DTZ over bis-CTZ.

Following the reviewer's suggestion, we have now included a synthetic analog 8pyDTZ that differs only one nitrogen atom from DTZ in **Fig. 4a-c**. LuxSit-i still exhibited 28-fold selectivity for DTZ over 8pyDTZ. The nitrogen on the C-8 substitution likely resulted in an energetically unfavorable buried polar atom impeding substrate binding.

Referee #2

This manuscript describes a new, partly computational approach, based on a combination of deep learning and more 'classical' physics-based optimization, to design enzymes with optimized substrate selectivities. The authors apply the approach to NTF2 proteins to design an optimized luciferase.

I was asked to focus on the deep learning component of the manuscript and so I will restrict my comments accordingly. On the whole, the approach described here does not deviate substantially from the trRosetta-based hallucination approach that was previously published (<https://www.nature.com/articles/s41586-021-04184-w>), including the use of KL divergence and MCMC search. The main innovation is a new fold-specific loss function that incentivizes the model to focus on NTF2-like topologies, by penalizing it when its hallucinations deviate from the fold. This is an interesting idea and was applied effectively in this work. On its own the novelty presented by this change is somewhat limited, but of course this is only one component of the overall manuscript.

We thank Reviewer #2 for the comments.

My main specific comments center around the description of the method, specifically section 3.2 in Methods, which is rather sparse and can use substantial beefing up. In particular:

- The sentence "For each seed structure, we used a structure-based sequence alignment 35 method (see 3.3) to find equivalent positions between that and a set of crystal structures (χ) trimmed to only include "ideal," designable elements." refers to ideal and designable elements. How are these formally defined? Right now the description is entirely qualitative and it's unclear how these elements are actually selected.

These elements were selected qualitatively, and we provide the trimmed set in the supplementary files. We revised the method sections to make this more clear:

"As many experimentally characterized NTF2s contain non-ideal regions, we began by creating a set (χ) of trimmed but ideal NTF2s by manually removing non-ideal structural elements such as kinked helices, and long or rarely observed loops. For each seed structure, we then used a structure-based sequence alignment method (see 3.3) to find equivalent positions between the seed structure and χ . Residue pairs were considered to be in a conserved tertiary motif (TERM) if there were 5 or more equivalent positions in χ ."

- The sentence "For pairs of positions with at least 5 crystal structure equivalent positions, smooth probability distributions were computed by applying a Gaussian blur to distances, a von Mises blur along omega angle, and von Mises-Fisher blur along phi and theta angles." gives high-level detail to how the loss function is constructed but no numerical details, making this irreproducible. The actual functional form, including hyperparameter choices, should be given.

We revised the method sections by adding: "The smooth probability distributions based on observed geometries in χ were then computed. For distances we used a Gaussian distribution with mean equal to the true distance denoted by D and standard deviation denoted by σ equal to 0.5 Å. The probability density function for distances d is given by:

$$f(d; D, \sigma) = \frac{1}{\sqrt{2\pi\sigma^2} \exp\left(-\frac{(d-D)^2}{2\sigma^2}\right)}$$

Using this density function one can construct a categorical distribution for binned distances by evaluating this function at the centers of the bins and then normalizing by a sum of all values in different bins. Similarly, a von Mises distribution was used for omega angle smoothing with probability density function given by $f(\omega; \Omega, \kappa) = N(\kappa) \exp[\kappa \cos(\omega - \Omega)]$ where $N(\kappa)$ is a normalizing constant, Ω is the crystal value, κ is the inverse variance chosen to be 100, and ω is the smoothed angle. For phi and theta angles a von Mises-Fisher blur is given by $f(x; \mu, \kappa) = N(\kappa) \exp[\kappa \mu^T x]$ where $N(\kappa)$ is a normalizing constant, μ is a unit vector on a 3D sphere corresponding to the phi and theta angles from the crystal structure, x is a smoothed unit vector, and κ is the inverse variance chosen to be 100. "

- This sentence, "Finally, we converted this to an energy by taking the negative log-likelihood 40 and calculated the expected energy given the trRosetta probability distribution to encourage the network to seek out the consensus structure, while still allowing deviations where needed." comes across as very vague. I really don't understand in detail what is happening here. Equations and/or pseudocode would communicate this much more easily. Right now the exposition feels like a high-level overview of what was done with few details on the actual implementation.

We added the following text in the method sections to describe the new loss function in detail.

"Next, we converted those probability distributions to energy landscapes (ie - negative log likelihoods) and sought to minimize the expected energy. This soft restraint encouraged the network to seek out the consensus structure, while still allowing deviations where needed. Specifically, we formulated the fold-specific loss as:

$$L_{fold} = \sum_{x \in \{d, \omega, \theta, \phi\}} \left[\sum_{i,j=1}^L \sum_{k=1}^{N_x} -m_{ij} p_{x,ijk} \ln(s_{x,ijk}) \right] / \sum_{i,j=1}^L m_{ij}$$

$$m_{ij} = \{ 1 \text{ if } i \text{ and } j \text{ are in a TERM; else } 0$$

where p is the network prediction and s is the smoothed probability distribution of the conserved residue pairs. For the second part of the loss function and similar to previous work¹⁹, we sought

to maximize the Kullback–Leibler (KL) divergence between the predicted probability distribution and a background distribution for all i, j residue pairs not in a TERM.

$$L_{hall} = - \sum_{x \in \{d, \omega, \theta, \phi\}} \left[\sum_{i,j=1}^L \sum_{k=1}^{N_x} (1 - m_{ij}) p_{x,ijk} \ln(p_{x,ijk}/b_{x,ijk}) \right] / \sum_{i,j=1}^L (1 - m_{ij})$$

where b is the background distribution and N_x is the number of bins in each probability distribution ($N_d = 37$, $N_{\omega, \theta} = 25$, $N_{\phi} = 13$). Briefly, b is calculated by a network of similar architecture to trRosetta trained on the same training data, except it is never given sequence information as an input. The final loss is given by:

$$L = L_{fold} + L_{hall}$$

“

- In terms of the four different kinds of MCMC moves, are they all equally likely?

We revised the method sections to describe this by adding: “We allowed four types of moves with different sampling probabilities: mutations ($p=0.55$), insertions ($p=0.15$), deletions ($p=0.15$), and moving segments ($p=0.15$).”

Finally, a code release would ameliorate the ambiguities described above by actually providing the tools to reproduce the computational experiments. Given the excellent standards set by the DeepMind team with AlphaFold2 and their code release, I strongly advocate that a similar code release be made here, with the full protocols needed, including Rosetta-based ones, to go from the initial sequence search for NTF2 homologs to the final designed structures.

All the scripts will be available on GitHub once published. At this stage, we provide the scripts from family-wide hallucination scaffolds generation to RosettaDesign via google drive (more RosettaDesign details in the supplementary):

<https://drive.google.com/file/d/1G8y0BqHG4wbdS2zjUeIOWOGVngPw1by/view?usp=sharing>

Referee #3

Baker and colleagues report the de novo design of a new luciferase enzyme. Computational approaches were used to identify candidate protein structures that could accept a “ready-to-react” luciferin analog and use it to generate light. A set of designer sequences was screened and a light emissive structure was identified. Further mutagenesis and biochemical assays were performed. The utility of the new luciferase was also evaluated in cultured cell models and in conjunction with other luciferin analogs. Overall, this work advances the field of protein design, making inroads into the design of enzyme catalysts—a formidable challenge. The experiments were well designed and executed. I am a little underwhelmed by current manuscript, though, based on the data provided and lack of discussion in key areas. My comments and concerns are outlined below.

We thank Reviewer #3 for the comments.

1. The manuscript is a little light on discussion regarding key design principles. When the NTF2-like superfamily is first mentioned (top of p. 2), the choice seems rather arbitrary. Several “pockets of appropriate size and shape” could exist. Perhaps the data in Fig. 1e-g are meant to provide additional rationale, but they are not discussed until later in the text. I found this section of the manuscript hard to navigate. What were the benchmarks for success? Were other protein scaffolds considered and/or attempted in this approach? Scaffolds that “didn’t work” might also be interesting from a mechanistic standpoint. Is there a way to compare the LuxSit sequence back to the original computational data set, in terms of relative ranking?

We searched a variety of folds before we chose to focus on NTF2-like superfamily, and have now clarified this in the text: “*To identify protein folds capable of hosting such pockets, we first docked DTZ into 4000 native small molecule binding proteins. We found that many NTF2 (nuclear transport factor 2)-like folds have binding pockets with appropriate shape-complementary and size for DTZ placement (red labels in Fig. 1e), and hence selected the NTF2-like superfamily as the target topology.*” The *in silico* benchmarks are based on RIF binding energy (Fig. 1e), which assesses the extent to which a scaffold can harbor a DTZ binding site, and AlphaFold2 predicted local distance difference test, pLDDT (Fig. 1g), which assesses how strongly the amino acid sequences encode their structures.

There are multiple factors that could make a design not work, such as poor protein folding, non-ideal catalytic geometry, the wrong polarity of the pocket, etc. With the current colony-based screening setup, we can only identify active luciferase designs. We might need to develop other high-throughput assays to distinguish and accumulate enough data to investigate the exact failure mode. What we can see from our current *in silico* scores of the three active DTZ designs we identified, is that their contact molecular surface and Rosetta ddG scores are all located at the upper end of the library distribution (see Fig. S1a), which suggests having substantial packing and contact with the ligand is an important factor for active luciferases.

2. Related to the above point, some additional details on the addition of a key Arg residue would be helpful. Is this residue also present in native luciferase structures (especially those that have been crystallized)?

To the best of our knowledge, Arg residue has not been proposed in any native luciferase to catalyze bioluminescence emission. The Apo-structure of engineered *Oplophorus* luciferase (5B0U) has an Arg residue in the putative pocket, but the catalytic mechanism of OLuc or its engineered nanoKaz/nanoLuc is still unclear. To support the Arg hypothesis, we performed molecular dynamics (MD) simulations, which suggest that the primary role of the designed Arg65 is to stabilize the charge on the high-energy transition states, TS1 and TS2. (**Extended Data Fig. 8b**).

It would also seem that if the luciferin anion is stabilized too much, turnover would be inhibited. Did this issue factor into the design?

Our initial hypothesis was that the first intermediate anion species was critical for catalysis of light emission. To investigate this, we performed independent density functional theory (DFT) calculations, which support the anion stabilization mechanism as the design principle (**Extended Data Fig. 8a** and **Fig. S2a**). It is not clear how much stabilization is needed to trigger the chemiluminescence of the substrate; thus, we sought to stabilize the anion with the proposed arginine and computationally sampled a range of local proton activity, solvent polarity, and hydrophobicity for the efficient activation of $^3\text{O}_2$ in the pocket (triplet oxygen can react directly with the anionic species of DTZ, **Extended Data Fig. 8a**). As we relied on experimentally screening

to identify active designs, we cannot rule out the possibility that some of the designs may bind to the substrate but do not have catalytic turnover.

Did the authors consider designing binders for a different luciferin intermediates along the reaction coordinate? Stabilized dioxetanes, etc. have been used in screens by others developing artificial luciferase-like enzymes and ribozymes.

This is an excellent suggestion. Our DFT-calculated energy profile for the mechanism of DTZ luminescence identifies anionic peroxide intermediates/transition states (**Extended Data Fig. 8a**) that could be design targets for future studies.

3. The manuscript hints at the generality of the approach, but this feature wasn't quite established. Luciferase appears to be a great initial choice, given that the only other requirement beyond the luciferin is molecular oxygen.

In this revision, by using the same design principle and methodology along with a deep-learning based protein sequence design method, ProteinMPNN, we've successfully designed de novo luciferases for another synthetic substrate, h-CTZ. The success rate to achieve hits has been greatly improved from screening 7982 to just testing 46 sequences. The new de novo luciferases were highly soluble, monodisperse, and monomeric; meanwhile, their luciferase activities were at the same order of magnitude as LuxSit (**Extended Data Fig. 4**). One of them exhibited high substrate selectivity toward the designed luciferin substrate – h-CTZ (**Fig. 4c**). These additional data we believe support the generality of the approach.

4. While appropriate biochemical assays were performed, the mechanism of light emission is still speculative. It's unfortunate that a crystal structure wasn't obtained to provide additional evidence for luciferin binding, etc. Perhaps it is possible to tease out details on the importance of the anion, though, using a synthetic control compound?

Thank you for the suggestion. DTZ is a relatively new synthetic luciferin. To the best of our knowledge, its synthetic control analog has not been synthesized. Instead, we performed DFT calculations (**Extended Data Fig. 8a** and **Fig. S2a**) to provide additional evidence to the importance of anion species developed during the reaction. Our result indicated that the reaction of the anionic intermediate reactants is strongly favored over the neutral form, which supported our initial hypothesis of stabilizing the anion species for luminescence.

5. LuxSit appears to be a viable probe for imaging studies. It would be good to see the raw light outputs alongside RLuc, GLuc, and NLuc, though (versus just normalized intensities in some cases). The light output might be too weak to pair alongside some of the brighter probes in the field for simultaneous imaging. It would also be helpful to see the raw photon data for the analog comparisons. It's difficult to get a sense for the selectivity as plotted.

We included the raw light outputs of LuxSit-i, RLuc, GLuc, and NLuc in **Fig. S3**.

Reviewer Reports on the First Revision:

Referees' comments:

Referee #1:

The authors have addressed all of my comments/concerns with this revision. Also, they describe a new set of successful designs targeting an alternative substrate. These additional results further validate the design approaches employed here and raise the impact of this study.

Referee #2:

The authors have satisfactorily addressed my concerns.

Referee #3:

Overall, the authors did a reasonable job addressing the reviewer critiques. I would recommend some more nuanced comments regarding the size and applications of the new reporters, though, given recently published work from Ueda (2022) and others.

Author Rebuttals to First Revision:

A point-by-point response

Referee #1:

The authors have addressed all of my comments/concerns with this revision. Also, they describe a new set of successful designs targeting an alternative substrate. These additional results further validate the design approaches employed here and raise the impact of this study.

We thank the reviewer for the comment.

Referee #2:

The authors have satisfactorily addressed my concerns.

We thank the reviewer for the comment.

Referee #3:

Overall, the authors did a reasonable job addressing the reviewer critiques. I would recommend some more nuanced comments regarding the size and applications of the new reporters, though, given recently published work from Ueda (2022) and others.

We have added additional comments in the conclusion regarding the potential application of our de novo luciferases.